behaviour, evolution

animal signals, lizard communication, epidermal gland secretions, squamates, social evolution

**Author for correspondence:**
Simon Baeckens
e-mail: simon.baeckens@uantwerp.be

# Investment in chemical signalling glands facilitates the evolution of sociality in lizards

Simon Baeckens[1,2] and Martin J. Whiting[2]

[1]Functional Morphology Laboratory, Department of Biology, University of Antwerp, 2610 Wilrijk, Belgium
[2]Department of Biological Sciences, Macquarie University, Sydney, NSW 2109, Australia

  SB, 0000-0003-2189-9548; MJW, 0000-0002-4662-0227

The evolution of sociality and traits that correlate with, or predict, sociality, have been the focus of considerable recent study. In order to reduce the social conflict that ultimately comes with group living, and foster social tolerance, individuals need reliable information about group members and potential rivals. Chemical signals are one such source of information and are widely used in many animal taxa, including lizards. Here, we take a phylogenetic comparative approach to test the hypothesis that social grouping correlates with investment in chemical signalling. We used the presence of epidermal glands as a proxy of chemical investment and considered social grouping as the occurrence of social groups containing both adults and juveniles. Based on a dataset of 911 lizard species, our models strongly supported correlated evolution between social grouping and chemical signalling glands. The rate of transition towards social grouping from a background of 'epidermal glands present' was an order of a magnitude higher than from a background of 'no epidermal glands'. Our results highlight the potential importance of chemical signalling during the evolution of sociality and the need for more focused studies on the role of chemical communication in facilitating information transfer about individual and group identity, and ameliorating social conflict.

## 1. Introduction

Amoebas, slime mould, ants, snapping shrimp, meerkats, chimpanzees and wolves all have one thing in common—they cooperate [1]. Group living in many species has set the stage for the evolution of complex sociality, including family living and even eusociality. Why sociality has evolved in some animal lineages, and not others, has been the subject of profound interest to biologists for decades [1]. Yet, it is only with the advent of modern phylogenetic comparative methods and a wealth of information on phylogenetic relationships in diverse taxa, that researchers have been well positioned to tackle bigger questions about social evolution [2]. For example, much attention has been focussed on the life-history and ecological correlates of sociality, including age at first reproduction, fecundity, dispersal, longevity [3–5], foraging mode [6], predation pressure [7] and the climatic environment [8–10]. Also, cognition [5], (allo)maternal care [11,12] and even reproductive mode [13] have been examined in relation to sociality. Nevertheless, we still have a poor understanding of the role of information transfer in driving social evolution, particularly in vertebrates. This is remarkable because all forms of social aggregations are reliant upon efficient communication for social recognition (e.g. individual and kin recognition) and the coordination of group behaviour or collective action [5,14]. Male sociality in bats, for instance, is believed to have evolved to promote information transfer about food [15]. Most of our knowledge on the topic, however, originates from work on insect societies and how an increase in pheromone complexity has facilitated the evolution of higher levels of sociality in insects

[16,17]. To the best of our knowledge, whether the mode of communication has influenced social evolution in terrestrial vertebrates remains largely unexplored (but see [18]).

Multi-modal signalling is the norm among animals, although particular modes may be more important in specific environments and contexts because they vary in detectability, speed of transmission and persistence in the environment [19–21]. To illustrate, in contrast with visual and acoustic signals, chemicals travel more slowly, last longer and vary greatly in spatial transmission depending on local environmental conditions (e.g. wind, substrate, temperature) [19,22,23]. Acoustic and visual signals are typically most effective for long-range communication [21,24], whereas chemical signals work optimally (and are more reliable) at relatively short distances (through airborne odours) or via direct contact (at least in terrestrial vertebrates [22,25]). Here, we hypothesize that reliance upon chemical signals for intraspecific communication increases the opportunity for short-range interactions between conspecifics (allowing efficient signalling), thereby creating conditions conducive for the transition from solitary to group living. We believe that lizards are a promising group to test this idea for at least three reasons.

First, some form of social aggregation has been reported for over 90 different lizard species in 22 different families [26–28]. Of these, at least 18 species from seven different families exhibit stable aggregations across years [28]. Social grouping has independently evolved multiple times in different lizard lineages [13]. Second, the sensory modalities through which lizards communicate vary greatly among taxa. While some lizards are known to use both visual and chemical signals to communicate with others [29–32], many species may depend more heavily on one mode. Specifically, some lizard clades such as the agamids are believed to be more 'visually oriented' while others, such as the lacertids, are thought to be more 'chemically oriented' [33,34], although we need more empirical support of these ideas. In the case of lizards that use chemical communication, several possible chemical sources may be responsible for signal production. While there is some evidence that cloacal exudates and faecal pellets may contain socially relevant information, particularly the skin and the epidermal glands (generation and follicular) are considered important sources of chemical signals in lizard communication, enabling mate assessment, individual recognition, species recognition and sex identification (reviewed in [35–39]). *Liolaemus tenius* females, for instance, are more attracted to substrates covered with male epidermal gland secretions than to substrates scent-marked with male skin extracts [40]. The occurrence (absence/presence) of lizard epidermal glands is strictly species-specific and regularly used as a proxy of a species' investment and reliance upon chemical signalling [35,41–45]. Lizards sample substrate-bound or airborne chemicals in the environment using tongue-flicking (vomerolfaction), or receive chemical-laden air through the nasal nares (olfaction *s.s.*) [46,47]. Third, with nearly 7000 lizard species and with well-supported phylogenetic relationships among species [48], phylogenetic comparative methods can be used to study correlated evolution between sociality and chemical signal investment. Here, we test the hypothesis that social grouping in lizards has evolved more readily from a state of high chemical signalling investment (the presence of epidermal glands) than from a state of low chemical signalling investment (the absence of epidermal glands). We predicted that species

with some form of sociality, specifically intergenerational groupings, were more likely to have invested in chemical signalling glands in their evolutionary history. Intergenerational social grouping (hereafter social grouping) is here (as in [13]) defined as the occurrence of social groupings containing both juveniles and adults.

## 2. Methods

### (a) Data collection

Data on lizard sociality were retrieved from Halliwell *et al*. [13], which performed a comprehensive phylogenetic comparative analysis of social grouping behaviour in squamate reptiles. We extracted information on the presence of intergenerational social grouping (i.e. between both adults and juveniles) for 911 lizard species belonging to 31 families. Only species represented in the phylogeny of squamates proposed by Pyron *et al*. [48] were included in this study. We refer to Halliwell *et al*. [13] for details on the exact definition of social grouping and the specific conditions used to code it as 'present' or 'absent'. Briefly, they included in their analysis any species for which there was a reported association between adults and juveniles that were indicative of social tolerance. This is important because social tolerance is the basis for parental care in lizards and the precursor for more complex parental care [26,27]. For two species (*Underwoodisaurus milii* and *Christinus marmoratus*), we were made aware that they do not conform to the social classification of Halliwell *et al*. [13] (based on [49]) and therefore did not score them as species with intergenerational social grouping in our dataset. A small part of the data collected by Halliwell *et al*. [13] was not retrieved from primarily literature, but from reports of observations by trained herpetologists. While we recognize its limitations, the data amassed by Halliwell *et al*. [13] is the most exhaustive dataset on lizard sociality to date. Next, we searched the literature for species-specific information on the absence and presence of epidermal (generation or follicular) glands in the femoral or (pre)cloacal region of lizards. We were able to gather epidermal gland data for all 911 species; the majority of gland data (approximately 95%) was retrieved from [35] and [42]. For every species, we thus contained binary data on the absence (0) or presence (1) of 'social grouping' (SG) and 'epidermal glands' (EG).

### (b) Data analysis

We began by pruning the phylogenetic tree [48] to include only the 911 species implemented in this study. Next, we tested for phylogenetic signal in the two variables 'social grouping' and 'epidermal glands' by calculating $D$ (a measure of phylogenetic signal in a binary trait [50]) using the function 'phylo.d' (1000 permutations) in the *caper* package [51].

We used three different statistical approaches to examine the role of epidermal glands in the evolution of social grouping. First, we used basic Markov models to test for correlated evolution between social grouping and epidermal glands [52,53]. Using the function 'fitPagel' (package *phytools* [54]), we fitted and compared four different models, each describing a different evolutionary scenario: (i) independent evolution of social grouping and epidermal glands, (ii) correlated evolution of social grouping and epidermal glands, (iii) evolutionary change in social grouping depends upon the state of epidermal glands, and (iv) evolutionary change in epidermal glands depends upon the state of social grouping. Second, to account for the potential joint effects of social grouping and epidermal gland on speciation and extinction rates, we used multi-state speciation and extinction modelling for discrete character evolution ('MuSSE' function in *diversitree* package [55]). To do so,

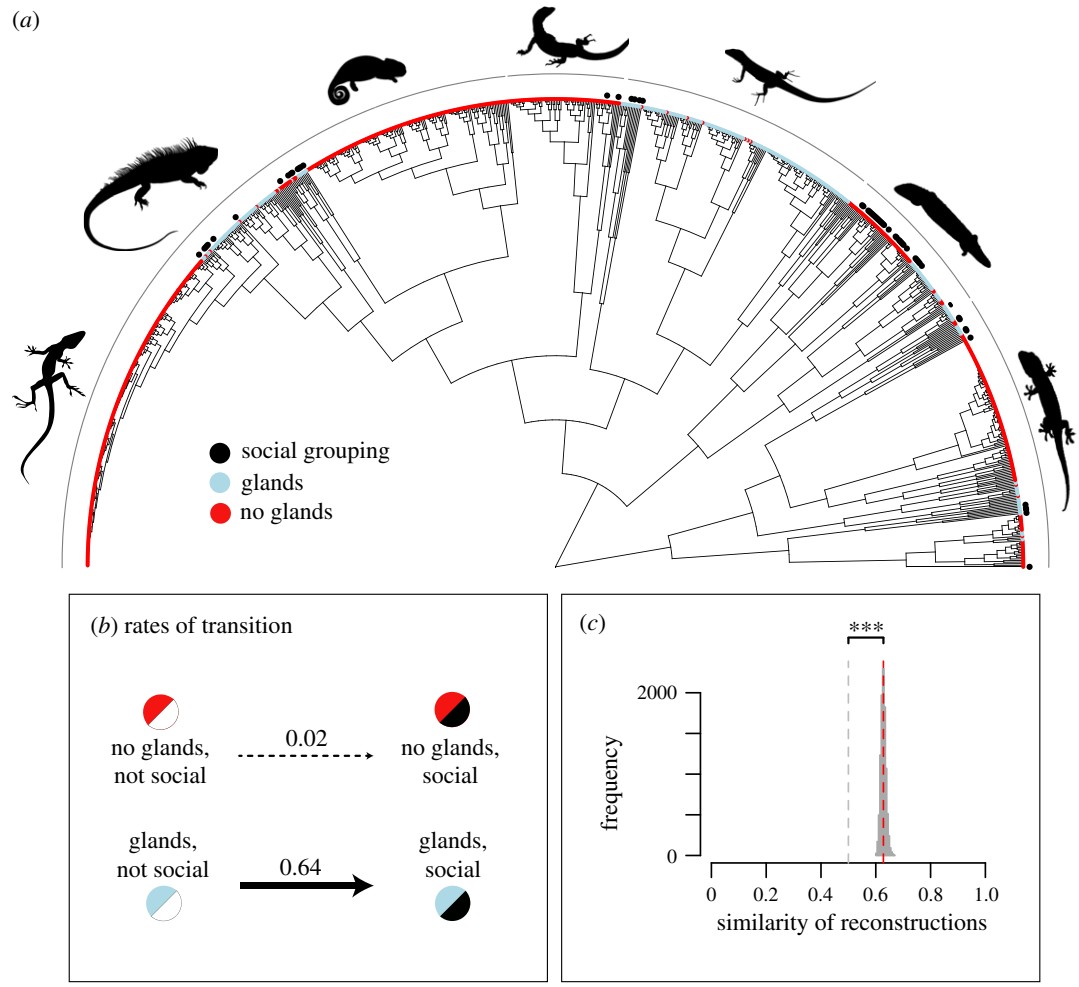

**Figure 1.** Coevolution of social grouping and epidermal glands in lizards. (*a*) Phylogenetic relationships [48] among the 911 lizard species included in our study with the colours at the tree tips indicating the absence (red) or presence (blue) of epidermal glands. Additional black dots denote the presence of social grouping in a species. (*b*) Rates of transition estimated using MuSSE modelling; the transition rate to social grouping from a state of 'epidermal glands present' is higher than from a state of 'epidermal glands absent' (table 2). (*c*) Histogram showing the fraction of a stochastic character mapping that is shared between the reconstructed trees of social grouping and epidermal glands. The mean similarity (0.63) is significantly higher than 0.5, indicating correlated evolution between social grouping and epidermal glands. Asterisks denote $p < 0.001$.

we first fitted a model that allowed transition rates from a state of 'social grouping absent' to a state of 'social grouping present' to vary depending on epidermal glands (here 'unconstrained model'). Next, a constrained version of the model was fitted in which the transition rates of 'social grouping absent' to 'social grouping present' was forced to be equal (constrained model). We subsequently compared the likelihood of both models using a chi-square test to examine if transitions to social grouping have occurred at different rates in lineages with or without epidermal glands. A second test was performed to explore the alternative (inverse) prediction (i.e. social grouping facilitates the evolution of epidermal glands); here, the transition rates of 'epidermal gland absent' to 'epidermal gland present' in the constrained model was forced to be equal. In the third approach, we performed ancestral state reconstructions via stochastic character mapping [56,57] to assess the strength of correlated evolution between social grouping and epidermal glands. The function 'make.simmap' (*phytools* package [54,58]) was used to fit a continuous-time reversible Markov model for the evolution of each of our two binary traits to the tree, and to generate a set of 100 stochastic maps for each character conditioned on the model fit and tip states. The similarity of reconstructions between map sets was estimated by computing the mean and distribution of overlap in stochastic maps. A *t*-test was used to assess the difference between the distribution of overlap in the map sets against an expectation of 0.5.

## 3. Results

Of the 911 lizard species included in this study, 291 species (31.9%) had epidermal glands. Sixty-three species had social grouping (6.9%), 27 (42.9%) of which have epidermal glands (figure 1*a*; electronic supplementary material, Data S1). Tests for phylogenetic signal in both traits revealed negative *D* values (social grouping (SG) = −0.12; epidermal glands (EG) = −0.45), which significantly differed from 1 (SG, $p < 0.001$; EG, $p < 0.001$), but not from 0 (SG, $p < 0.705$; EG, $p = 0.991$), indicating non-random association of social grouping and epidermal glands with respect to the phylogeny.

Basic Markov models strongly supported the scenario of correlated evolution between social grouping and epidermal glands, in which the evolution of social grouping depends on the presence of epidermal glands (table 1). Similar results were obtained from the more advanced MuSSE models that accounted for variation in speciation and extinction rates associated with each character state. The 'unconstrained model' of the MuSSE analysis was favoured over the 'constrained model' indicating that transitions to social grouping occurred at different rates in lineages with and without epidermal glands ($\Delta$AIC = 8.80, $\chi^2 = 10.77$, $p = 0.001$; table 2). More specifically, the rate of transition to social grouping from a

**Table 1.** Performance of Markov models for the hypothesis of evolutionary association between social grouping (SG) and epidermal glands (EG). (Dependent evolution models were compared to the null model (independent evolution) using log-likelihood (LogL) ratio tests.)

| model | | LogL | AIC | *LL*-ratio | *p*-value |
|---|---|---|---|---|---|
| independent evolution | | −244.233 | 496.466 | | |
| dependent evolution | correlated evolution | −237.787 | 490.573 | 12.892 | 0.012 |
| | change in SG depends upon state of EG | −240.108 | 492.216 | 8.250 | 0.016 |
| | change in EG depends upon state of SG | −243.032 | 498.065 | 2.401 | 0.301 |

state of epidermal glands 'present' (rate = 0.635) was approximately 40 times higher than from a state of epidermal glands 'absent' (rate = 0.015) (figure 1*b*; table 2). We found no statistical support for the inverse prediction, i.e. that transitions to epidermal glands occurred at different rates in lineages with and without social grouping ($\Delta$AIC = 2.00, $\chi^2 < 0.01$, $p = 0.995$). Furthermore, transition rate towards social grouping from a background of epidermal glands was significantly higher than transition rate to epidermal glands from a background of social grouping ($\Delta$AIC = 3.60, $\chi^2 = 5.57$, $p = 0.018$). These findings of correlated evolution between epidermal glands and social grouping were also corroborated by the third approach, which revealed a mean similarity of 0.63 (standard deviation = 0.01) between stochastic character map sets based on separate ancestral character state reconstructions of social grouping and epidermal glands (figure 1*c*). Mean similarity differed significantly from the null expectation ($H_0 = 0.5$) of independent evolution between social grouping and epidermal glands ($t_{9999} = 1517.6$, $p < 0.001$).

## 4. Discussion

Our phylogenetic comparative analyses indicate that the presence of epidermal glands facilitated the emergence of social groupings (parents and offspring) throughout lizard evolutionary history. These social associations evolved at a considerably higher rate from a state where chemical signalling glands were present, than from a state in which they were absent. This suggests that the use of chemical signals may have promoted selective conditions for the evolution of sociality such as prolonged parent–offspring associations. Although one cannot infer causality from our methods (any unconsidered trait that is correlated with the target trait could be causal [55,59,60]), our findings highlight the potential important role of chemical communication in the evolution of lizard sociality.

How do chemical signals facilitate the emergence and evolution of social groupings and complex sociality? The most obvious explanation is that chemical communication facilitates recognition and bonding among parents and offspring. Kin are expected to be in close spatial proximity following birth. Furthermore, because sociality is closely associated with viviparity [13], offspring are expected to be in close association with their mothers and potentially also their fathers in pair-bonded species. This social setting and proximity sets the stage for easy access to chemical information. Not only will offspring directly sample chemical information from siblings, parents and relatives, but they will indirectly sample this same information from the surrounding environment by, for example, tongue-flicking the

substrate or refuge. This combined sampling behaviour probably provides a feedback loop and helps reinforce these associations and promotes the emergence and refinement of parent–offspring and kin-based sociality.

Efficient communication among individuals is fundamental for social recognition and—by extension—the formation and maintenance of social aggregations [14]. In lizards, chemical signals are a prime candidate medium to do so, for at least two reasons. First, reptilian chemical signals are composed of complex mixtures that carry detailed information of the emitter [25,61], which permits individual recognition [38,62] and may inform about their physiological state. Indeed, chemical signals from epidermal glandular secretions enable lizards to discriminate between con- and heterospecifics [63], familiar and unfamiliar conspecifics [64], males and females [65], young and old males [66], low and high parasite-infected individuals [67], and between conspecifics with a vitamin-poor and–rich diets [68], to name a few. Second, and unlike visual and acoustic modalities, chemical signals such as scent-marks work in darkness and around obstacles, operate in the absence of the signaller and persist for long periods of time in the environment (e.g. up to months for the snake *Crotalus viridis* [69]). Scent-marks effectively transmit the same information both passively and continuously, thereby increasing the probability of detection and accurate processing by conspecifics [19,23] (but also heterospecifics [70,71]) and help stabilize social systems [72–74].

A direct consequence of their physical properties, however, is that scent-marks have the disadvantage of being poor long-distance signals because they travel slowly and lack directionality; therefore, the efficient use of scent-marks is largely restricted to close-range communication [22,25,75]. Lizards cope with these constraints by closely approaching scent-marks and using tongue-flicking behaviour to obtain information content [76]. For example, short-distance chemical sampling enables flat lizard males (*Platysaurus broadleyi*) to correctly discriminate between females and female-mimicking males. In this species, males misidentify female mimics at a distance and court them using visual cues, but once they have physical contact via tongue-flicking, they are able to correctly identify their male sex [31]. In iguanas (*Iguana iguana*), the low-volatile fraction of secretion deposits holds detailed information on the identity of the scent-marker but needs to be extracted by tongue touches [64]. While it is probable that efficient chemical communication indirectly generates conditions conductive to selection for social aggregations, the exact mechanism linking chemical signalling and the emergence of animal sociality requires further examination.

Here, we found strong evidence for correlated evolution between social grouping and chemical signalling in lizards

**Table 2.** Performance and comparison of the MuSSE models to test the hypothesis that transitions to social grouping (SG) have occurred at different rates in lineages with and without epidermal glands (EG). (In contrast to the 'unconstrained model', transition rates from 'SG absent' to 'SG present' were forced to be equal in the constrained model. Simultaneous double transitions were prohibited in both models. Speciation ($\lambda$), extinction ($\mu$) and transition ($q$) rates are shown for species with: $a$, SG absent and EG absent; $b$, SG present and EG absent; $c$, SG absent and EG present; and $d$, SG present and EG present. A chi-squared ($\chi^2$) test compared the log-likelihood ($LogL$) of both models.)

| model | $\lambda_a$ | $\lambda_b$ | $\lambda_c$ | $\lambda_d$ | $\mu_a$ | $\mu_b$ | $\mu_c$ | $\mu_d$ | $q_{a\rightarrow b}$ | $q_{c\rightarrow d}$ | d.f. | LogL | AIC | $\chi^2$ | $p$-value |
|---|---|---|---|---|---|---|---|---|---|---|---|---|---|---|---|
| unconstrained | 1.705 | 1.527 | 2.121 | <0.001 | <0.001 | <0.001 | <0.001 | <0.001 | 0.015 | 0.635 | 16 | −670.13 | 1372.2 | | |
| constrained | 1.742 | 0.845 | 1.991 | 0.980 | <0.001 | <0.001 | <0.001 | <0.001 | 0.068 | 0.068 | 15 | −675.51 | 1381.0 | 10.77 | 0.001 |

using the presence of glandular or follicular epidermal glands as a proxy of chemical investment. Over the last two decades, studies of natural products chemistry combined with comprehensive behavioural assays have shown that the waxy secretions produced by epidermal glands are an important source of chemical signals for lizard chemical communication in a range of different clades [35,36,40,42]. Based on the most comprehensive literature search to date ($n = 4341$ lizard species), approximately 25% of all lizard species are equipped with follicular epidermal glands [42]. These structures were probably absent in the lizard common ancestor, and repeatedly emerged and disappeared in squamate evolutionary history [35,42]. Because of their significance in chemical signalling, the absence of epidermal glands in lizards has been interpreted by some researchers as less reliance on chemical signals as an information source [35,41–45]. A recent behavioural study showed that *Liolaemus* species with epidermal glands relied more upon chemical, rather than visual, signals for species recognition in contrast to species lacking glands, which appeared to rely predominantly on visual cues or signals for discriminating conspecifics and heterospecifics [77].

It is important to note that epidermal gland secretions are not the sole source of chemical information in lizards; faeces, cloacal secretions and skin lipids can contain socially relevant chemical stimuli too [36,39]. In other words, species that lack glands are not necessarily constrained by their ability to obtain chemical information [78]. For example, some lizard species 'scat pile', where they essentially have a latrine near their refuges and these act as information centres [79,80]. The family-living Egernia 'group' clade [81], for instance, lack epidermal glands, but tongue-flick assays indicate that they are able to recognize kin solely based on the scent (often of scats) alone [79,82–86]. Furthermore, although there can be significant sibling conflict among Egernia group lizards when litters are raised in the laboratory, family groups are stable and do not exhibit overt within-group aggression in the wild. In these family groups, individuals all bask in close proximity to one another and there is evidence of parental care [26,27,87]. One reason for this stability is the likelihood that members are able to recognize one another, and chemical cues could be integral to facilitating recognition while reducing conflict and increasing cooperation. While reports on the chemical abilities of the highly social Egernia group lizards essentially strengthen and validate the main findings of our study, they also highlight the shortcomings of classifying species based on the occurrence of epidermal signalling glands. Other work suggests that some species rely on a combination of chemicals from different sources to enhance signal effectiveness or to broadcast different messages at once [40,88]. Skin extracts and epidermal gland secretions of male *Liolaemus tenius* lizards, for instance, are found to mainly trigger marking behaviour in female conspecifics, while the scent of their faeces elicited predominantly display behaviour [40]. Exploiting multiple chemical sources may be particularly useful in lizard species for which active epidermal glands are restricted to a single-sex only. In species where females lack active epidermal glands, such as *Podarcis hispanicus*, male conspecifics can extract information from female cloacal and body odours to determine, for instance, reproductive condition [89], while females can profit from the information-rich epidermal gland secretions to assess, for example, a male's immune response [90]. It would be interesting to examine chemically mediated social tolerance among adults and young in species with females lacking

active epidermal glands. Also, chemical communication in the offspring-to-parent direction may involve multi-source chemical signalling, notably in species where glandular secretions only start to differentiate at the onset of sexual maturity. In such cases, parents probably rely on a mixture of chemicals from various sources for recognizing kin [91,92]. The diversity of chemical sources involved in lizard communication and the inter- and intraspecific variation in the importance of each source in producing socially relevant signals illustrates the imperfections of epidermal glands as a measure for chemical signalling investment.

An alternative proxy is to gauge a species' reliance upon chemical communication using the 'baseline tongue-flick rate', which, reflects a species' fundamental level of chemosensory investigation via lingual sampling for vomeronasal analysis [93]. Unfortunately, such information is only available for a limited number of species and is highly subjective to deviations in experimental study design [76,93]. This makes baseline tongue-flick rate a rather poor variable for comparative and macro-evolutionary analyses. Furthermore, lizards tongue-flick not only to sample conspecific scent-marks, but also to find food and detect predators. The relationship between tongue-flick behaviour and other confounding factors, such as foraging mode and diet [93–95], probably impedes an accurate examination of co-evolutionary patterns between tongue-flick behaviour and lizard sociality. Ideally, one would possess detailed information on the composition, diversity and richness of species' chemical signals (as in [96,97]). This would allow tests of coevolution between signal complexity and the level of social organization ('social complexity hypothesis for communicate complexity'; [98,99]) as shown in halictid bees [100] and strepsirrhine primates [101,102]. Alas, most of our (limited) knowledge of lizard chemical signal design originates from a few lizard taxa only [35,36], covering little variation in the degree of social organization; such clustered phylogenetic sampling constrains, at this point, reliable coevolutionary diversification analyses of chemical signal and social complexity in lizards. Irrespective of the challenges associated with the use of epidermal glands as a proxy for investment in chemical signalling, we show that the presence of epidermal glands facilitated the emergence and evolution of lizard sociality most commonly in the form of simple parent–offspring associations. This finding highlights the potentially important role of chemical communication, which is relatively cryptic, as an underappreciated mechanism for mediating lizard social evolution.

Data accessibility. Data are deposited at the Dryad Digital Repository: https://doi.org/10.5061/dryad.3xsj3txfg [103].

Authors' contributions. S.B. and M.J.W. conceived the study; S.B. gathered and analysed the data, and S.B. and M.J.W. wrote the manuscript.

Competing interests. The authors declare no competing interests.

Funding. S.B. was supported by the Research Foundation-Flanders (FWO) (12I8819N; V427219N)

Acknowledgements. We thank G.M. While for insightful discussions and three anonymous reviewers for constructive feedback on an earlier version of the manuscript.

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
