## [Peer Review File · Proceedings of the Royal Society B: Biological Sciences]

Review History

RSPB-2020-2438.R0 (Original submission)

Review form: Reviewer 1 (Anthony Herrel)

Recommendation

Accept with minor revision (please list in comments)

Scientific importance: Is the manuscript an original and important contribution to its field?

Good

General interest: Is the paper of sufficient general interest?

Good

Quality of the paper: Is the overall quality of the paper suitable?

Good

Is the length of the paper justified?

Yes

Should the paper be seen by a specialist statistical reviewer?

No

Do you have any concerns about statistical analyses in this paper? If so, please specify them explicitly in your report.

No

It is a condition of publication that authors make their supporting data, code and materials available - either as supplementary material or hosted in an external repository. Please rate, if applicable, the supporting data on the following criteria.

Is it accessible?

Yes

Is it clear?

Yes

Is it adequate?

Yes

Do you have any ethical concerns with this paper?

No

Comments to the Author

I have now read the paper entitled 'Investment in chemical signalling facilitates the evolution of sociality in lizards' submitted by Baeckens and Whiting. I found the question interesting and the analyses and conclusions compelling. The analyses are appropriate and interpreted carefully. I only have minor comments for the authors as listed below.

line 36: of 910 lizard species

line 39: rate ... 15 time higher (rates are not likelihoods)

line 70: add a space before the reference number

lines 112-114: lizards also use olfaction, yet this remains dramatically understudied but should be mentioned. Especially in geckos this may be an important sense.

line 131: why use the Pyron et al phylogeny and not some of the more recent properly time-calibrated and more complete phylogenies that are out there?

line 146: briefly explain 'D' here as it is the first time it is mentioned.

line 193: findings of correlated evolution ...

line 204: throughout lizard evolutionary history.

line 207: I would delete 'favourable'

lines 239-242: yet scent marks offer less flexibility ... they stay longer and also provide excellent cues for predators like snakes. So they also have some disadvantages.

line 267: I was a little confused here ... you say 25% and list 4341 species ... is that squamates or lizards ? please clarify.

line 289: one another, and ...

line 299: study design [67,81]. This makes ...

figure 1: great figure ! the circles above the phylogeny are a little small for old people like me so me make them a tiny bit larger ?

Review form: Reviewer 2

Recommendation

Major revision is needed (please make suggestions in comments)

Scientific importance: Is the manuscript an original and important contribution to its field?

Good

General interest: Is the paper of sufficient general interest?

Excellent

Quality of the paper: Is the overall quality of the paper suitable?

Good

Is the length of the paper justified?

Yes

Should the paper be seen by a specialist statistical reviewer?

No

Do you have any concerns about statistical analyses in this paper? If so, please specify them explicitly in your report.

Yes

It is a condition of publication that authors make their supporting data, code and materials available - either as supplementary material or hosted in an external repository. Please rate, if applicable, the supporting data on the following criteria.

Is it accessible?

Yes

Is it clear?

Yes

Is it adequate?

No

Do you have any ethical concerns with this paper?

No

Comments to the Author

This study ("Investment in chemical signalling facilitates the evolution of sociality in lizards") tests the hypothesis that social grouping in lizards (defined as groups containing young and adults) is correlated with the presence of chemical signal-producing epidermal glands, which the authors use as a proxy to infer investment in chemical signalling. The authors used phylogenetic comparative methods to assess the correlation between social grouping and the presence of epidermal glands among 910 lizard species. The results from three different statistical approaches all indicate that social grouping is more likely to arise in lineages possessing epidermal glands,

supporting their hypothesis. I enjoyed reviewing this manuscript and think that, pending major revision, it is potentially a good fit for publication in *Proceedings B*. However, I have major concerns that I feel need to be addressed. These concerns revolve around how the study is framed, how results are interpreted, and errors and other aspects of the underlying data. I also have a few questions about the statistics used, but am pleased to say the approach seems good to me overall, although I admit I am not a specialist in these specific analyses.

I contest the claim that the presence of epidermal glands reflects investment in chemical signalling, and thus think that the main conclusion of this paper is overstated (although I find the results themselves nevertheless interesting). While the presence/absence of glands might reflect chemical signal investment among some closely related species (e.g., *Liolaemus*; Ruiz-Monachesi et al. 2020, *Amphibia-Reptilia*), skinks, snakes, varanids, and other diverse squamate groups use chemical signals (including in social interactions) and yet do not possess epidermal glands. You acknowledge exactly these points in the Discussion, which I was very pleased to see, but a reader would not get any impression of these caveats reading your title, abstract, or introduction. I think the paper (including the title) needs to be edited so that these are not conflated. See line number comments below for some specific examples.

This is a correlational study, as are almost all macroevolutionary studies. You are usually careful to phrase things so as not to conflate correlation with causation, with the major exception of your paper's title, and a few smaller instances (see line comments). You cannot be certain that the presence of epidermal glands is not correlated with another trait that is directly influencing the evolution of sociality. Perhaps sociality is actually less likely to evolve in shrub-dwelling lizards (e.g., *Anolis* and chameleons), which appear less likely to possess epidermal glands (Baeckens et al. 2015). This is just the first example that came to mind; I'm not suggesting that claim is likely true. Conflating correlation and causation is, unfortunately, extremely common among macroevolutionary papers, but is nevertheless worth avoiding.

For some species with both epidermal glands and social grouping, only males possess epidermal glands (e.g., *Naultinus*, *Hoplodactylus*, *Rhacodactylus*; I assume there are more), and evidence suggests that epidermal glands in lizards do not differentiate until the onset of sexual maturity (briefly reviewed in Mayerl et al. 2015, *Amphibia-Reptilia*). Assuming I'm not mistaken about these claims, how do you reconcile these observations with your results? This study specifically tests the correlation between epidermal glands and social grouping defined as groups containing adults and juveniles. How can epidermal glands mediate interactions among adults and juveniles if juveniles do not possess functioning glands, and if (for some species) only a single sex possesses them? This is another reason why I urge caution with inferring causation, and I think these points are worth discussing.

While going through the data I found some errors. Both *Amblyrhynchus cristatus* and *Brachylophus* (2 spp.) are listed as not possessing epidermal glands, but they do, with papers published on the composition of secretions from these glands in *Amblyrhynchus* (<https://peerj.com/articles/3689/>). These species are early on alphabetically and so caught my eye easily, but it's worth checking the data more thoroughly. (ref for *Brachylophus vitiensis*: <https://journals.plos.org/plosone/article?id=10.1371/journal.pone.0073127>).

I looked at the data and references from Halliwell et al., with a focus on species I'm most familiar with, just to get a personal impression of how accurate the claims of social grouping are. In your study and in Halliwell et al. 2017, social grouping is defined as: "social groups containing both adults and juveniles" (quoted from Results section of Halliwell et al. 2017). Considering this, I take issue with the classification of *Underwoodisaurus milii* and *Christinus marmoratus* as species with social grouping. Their classification as such is based on Kearney et al. 2001 (*Herpetologica*), which states for *U. milii* (as *Nephrurus milii*): "For example, adult females of *N. milii* were rarely found with juvenile conspecifics", and, "...members of an aggregation tended to resemble each other in body size". While rarely doesn't mean never, this does not support that this species has inter-generational social grouping. In that same study, there was evidence of

aggregations in *Christinus marmoratus*, with the main finding that groups tended to contain only a single male but otherwise reflected what would be expected by chance. I'm not contesting that these species aggregate, but there's not good evidence that there is social grouping as defined here. Considering that these species lack epidermal glands, changing their social grouping classification would make your results even stronger. But the other consequence of all this is that my confidence in the classifications made by Halliwell et al. is somewhat lower. Halliwell et al. classified many species based on a single book (Somma 2003), that for many species (e.g., *Gekko gecko*) cites mainly anecdotal observations in terraria. Because social grouping in lizards appears relatively rare, even a few misclassifications can bias the results of your statistical tests, and so it's worth worrying about and probably checking – and at the very least, acknowledging as a limitation.

Why were snakes excluded from your analyses? They were included in Halliwell et al. 2017, they demonstrate social grouping as it is treated herein, and they are simply a diverse radiation of reduced-limbed lizards. Other reduced-limbed radiations such as pygopodids and amphisbaenids are also included here, so I see no sound reason to exclude snakes from the analysis. That they do not possess epidermal glands would not be a good reason because other diverse clades lacking glands (and sociality; e.g., anoles & chameleons) are included.

According to Garcia-Roa et al. 2017, 25% of lizards have epidermal glands, but for the species included in your study, 31.7% have glands. Thus, your data are not a random sample of the phylogeny, and some groups are underrepresented (e.g., skinks). Could this sampling bias influence the results of your statistical analyses? I'm not familiar enough with the underlying assumptions of these analyses to make specific comments, but my impression is that – as with statistics generally – a non-random sample will yield spurious conclusions. If this is the case it should be discussed as a limitation of the study, and if it is not the case then I just need some reassurance. On a related note, there's contention about the validity of ancestral state reconstructions, with a recent study (Holland et al. 2020, Scientific Reports) indicating that they are often unreliable when traits are not selectively neutral. Notably, SSE models (including MuSSE, which was used here) are prone to high type I error rates when the trait does not influence speciation and/or extinction rates. How does MuSSE perform with your data compared to other models?

The caption for Figure 1 indicates that branches should be coloured according to the ancestral state reconstruction, but from what I can see all branches are black.

I would like to see all R code and a file containing the pruned phylogeny added to the electronic supplementary material for scrutiny and reproducibility.

LN27: I suggest striking this first sentence. Your study addresses the correlates of sociality, but not *why* organisms are social, and so it seems a little out of place – plus the next sentence works well as an opener in my opinion.

LN35: As per my earlier comment, you test the correlation with epidermal glands, not with investment in chemical signalling.

LN35–37: Here you say that 32% of 910 species have epidermal glands, and *of those*, 42% are social – but this directly contradicts your raw data. In fact, only 27 of the 288 gland-bearing species are considered social here, which is 9% of species with epidermal glands.

LN40: Conflating correlation with causation. Your results highlight the “potential” importance of chemical signalling via epidermal glands. I do share your hypothesis that chemical communication is likely very important for the evolution of sociality, but these results (if robust) are correlational.

LN60–61: I'm not a fan of this opening sentence. It comes off as a bit sensationalist, like the opening of a David Attenborough documentary. Also, to be a pedant, those organisms all have other things in common, too. This is just my superficial impression – I leave the writing style entirely to you.

LN62: Stray comma. I suggest the comma goes between “sociality” and “including”.

LN64: “Biologist” should be plural, “biologists”.

LN90: No need for the hyphen between “short-distances”.

LN92: You're not really arguing this, rather, you're proposing it as a hypothesis.

LN96: “...for, for...”

LN104–107: And yet agamids overwhelmingly possess epidermal glands, while skinks lack them entirely. This calls into question LN110–112, which posits that the presence of epidermal glands reflects investment in chemical signalling, even though you've just mentioned that skinks (which lack epidermal glands) are more chemically oriented. I realise you qualify this with the comment about needing more empirical support, but it seems clear that the presence/absence of epidermal glands is not a good indication of “investment” in chemical signalling. I think this is a misleading way to frame the problem, and leads to unfounded conclusions when interpreting your results. See my other comments on this matter.

LN113–114: There's evidence suggesting that geckos are olfactory specialists that do not rely heavily on tongue-flicking (Schwenk 1993, *J. Zoology*), although I acknowledge that we need a lot more data on this.

LN126: Consider adding the skink *Nangura spinosa* (sometimes *Concinnia spinosa*) to your analysis. It is in the phylogeny of Pyron et al. and has social grouping, although this species wasn't included in Halliwell et al., possibly because the references for it (see following ref) are mainly presented in government reports and field guides (e.g., Wilson & Swan 2017). Department of Environment and Resource Management 2010. Recovery plan for the Nangur spiny skink (*Nangura spinosa*). Report to the Department of Sustainability, Environment, Water, Population and Communities, Canberra. Department of Environment and Resource Management, Brisbane.

LN127: Should be “...phylogenetic comparative analysis...”

LN131: Needs a full stop in “al.”

LN132–135: I think the definition of social grouping used herein should be explicitly stated in the Introduction, and maybe in the Abstract if there's room for it. Social grouping can be conceptualised in various ways, so I think it's best to be as up-front about it as possible to avoid confusion (and just make it easier for the reader).

LN139: A lot of the classifications regarding epidermal glands comes from Mayerl et al. 2017. Is the data from this paper publicly available? I can't find it.

LN141: Does “epidermal glands” include generation glands, or only epidermal follicular pores?

LN144: In the text file with your raw data, the species column states names are from Pyron et al. 2011, but I assume you actually mean Pyron et al. 2013 [ref 44]?

LN165: I assume you're saying you used a chi-squared test? If so I suggest rephrasing to make more obvious, e.g., “We compared the likelihoods of these two models using a chi-squared

test...”

LN170: Can you explain this analysis better? I had to read Halliwell et al’s methods to understand what was going on here (assuming you did this the same way they did).

LN171: I’m unfamiliar with the details of make.simmap, but is there a reason the permutations are written as 100 x 100 rather than 10,000?

LN181: “...with respect to the phylogeny.”

LN197: I suggest striking “highly”.

LN199: Degrees of freedom are not meaningful for permutational analyses. The p-value is calculated from the permutation distribution.

LN203: Conflating correlation with causation. Remove “facilitated” and rephrase.

LN203–204: Where has this pair-bonded and kin-based stuff come from? The social grouping classification is based on groups consisting of adults and juveniles as per Halliwell et al. 2017, not pair-bonded adults or anything to do with kin (except in some of their stable social grouping stuff, but that’s not used here). This statement is beyond the scope of this study. Same comment goes for LN213–215. For many of the studies Halliwell et al. used to classify social grouping, there was actually no way to know whether the observed juveniles were actually the offspring of the adults they were found with, or kin.

LN212–225: What about the fact that for many species only males possess epidermal glands, and that juveniles of lizards possessing epidermal glands will not have fully formed and functional epidermal glands? How can epidermal glands then mediate social interactions in these cases?

LN209: It highlights the *potential* role.

LN226: Must communication really be continuous?

LN234: Regarding “juveniles and adults” – the cited study looks at female discrimination of “young” and “old” males, but there is no indication that these are sexually immature juveniles. In fact, the words “juvenile” and “immature” do not appear in the manuscript (except in the reference list).

LN262–266: These references do not demonstrate that epidermal glands are *the* main source of chemical signals in lizards, even if that claim is frequently made within them. Three of these are reviews (which of course do review how epidermal glands are an *important* source of chemical signals), and one is a macroevolutionary study (although Mayerl et al. also has a bit of macro in it). None of these references test that epidermal glands are behaviourally more important than other sources of chemical signals (e.g., urodaeal glands, faeces, etc.). If there are studies that support this claim broadly across lizards please cite them directly. I can’t help but get the impression that ref 68 is a gratuitous self-citation here.

LN270–272: This is false. We know that skinks, varanids, and snakes rely heavily on chemical signalling, and none of these possess epidermal glands. It might be true in specific circumstances (e.g., Ruiz-Monachesi et al. 2020), but is demonstrably false in broad terms. And you go on over the rest of the paragraph to acknowledge this – there are many sources of import, socially mediating chemical signals in lizards. Thus, is there any value in stating something like: “Because of their significance in chemical signalling, the absence of epidermal glands in lizards likewise suggests less reliance on chemical signals as an information source” before diving in to these caveats?

LN279: I suggest replacing “depauperated” with a more commonly used synonym, e.g., “reduced”.

LN294–305: You say tongue-flicking is more suitable in one sentence, then in the next sentence contradict yourself and say it’s too biased and flawed. You could simply say that baseline tongue-flick rates are an alternative proxy (without saying it’s probably better), and then outline its strengths and weaknesses without contradicting yourself.

LN305–309: You do not show any of this. You find a correlation between the presence of epidermal glands and the evolution of social grouping defined as associations between adults and juveniles. This is not evidence that epidermal glands facilitated the evolution of social grouping behaviour (conflating correlation and causation), and social grouping was not defined by pair-bonds or family units. Halliwell et al. give an additional and stricter classification of “stable social grouping”, but that is not what was analysed here.

LN327: Branch colours are not visible on the figure. All branches are black.

Review form: Reviewer 3

Recommendation

Accept with minor revision (please list in comments)

Scientific importance: Is the manuscript an original and important contribution to its field?

Acceptable

General interest: Is the paper of sufficient general interest?

Good

Quality of the paper: Is the overall quality of the paper suitable?

Good

Is the length of the paper justified?

Yes

Should the paper be seen by a specialist statistical reviewer?

Yes

Do you have any concerns about statistical analyses in this paper? If so, please specify them explicitly in your report.

Yes

It is a condition of publication that authors make their supporting data, code and materials available - either as supplementary material or hosted in an external repository. Please rate, if applicable, the supporting data on the following criteria.

Is it accessible?

Yes

Is it clear?

Yes

Is it adequate?

Yes

Do you have any ethical concerns with this paper?

No

Comments to the Author

The authors are suggesting that the mode of communication facilitates the occurrence of sociality, i.e. that the initial presence of glands allows and contributes to the development of sociality. However, in principle, one would expect that signal complexity increases as the need for communication increases. Or more likely that there is a coevolution process in which an initial low investment in communication increases to a higher level of investment as sociality develops. I was not convinced that these alternative scenarios were properly refuted or even discussed.

Minor comments

Abstract: I find this sentence in the Abstract confusing: "We found that roughly 32% of 910 species invest in signal-producing epidermal glands and of these, approximately 42% form social groups." This seems to imply that 58% of species with epidermal glands, i.e. a majority of them, do not form social groups, and this seems to contradict your main conclusion. People reading the abstract without having read the rest of the paper will get confused.

Lines 81-82: "Whether the mode of communication has influenced social evolution in terrestrial vertebrates remains largely unexplored." No references are given here, which overemphasizes the "unexplored" claim to a point that is not correct. For example, there are studies in strepsirrhine primates on the matter.

Line 96: correct "for, for"

Decision letter (RSPB-2020-2438.R0)

12-Nov-2020

Dear Dr Baeckens:

Your manuscript has now been peer reviewed and the reviews have been assessed by an Associate Editor. The reviewers' comments (not including confidential comments to the Editor) and the comments from the Associate Editor are included at the end of this email for your reference. As you will see, the reviewers and the Editors have raised some concerns with your manuscript and we would like to invite you to revise your manuscript to address them.

When submitting your revision please upload a file under "Response to Referees" - in the "File Upload" section. This should document, point by point, how you have responded to the reviewers' and Editors' comments, and the adjustments you have made to the manuscript. We

require a copy of the manuscript with revisions made since the previous version marked as 'tracked changes' to be included in the 'response to referees' document.

Research ethics:

Use of animals and field studies:

It is a condition of publication that you make available the data and research materials supporting the results in the article. Please see our Data Sharing Policies (<https://royalsociety.org/journals/authors/author-guidelines/#data>). Datasets should be deposited in an appropriate publicly available repository and details of the associated accession number, link or DOI to the datasets must be included in the Data Accessibility section of the article (<https://royalsociety.org/journals/ethics-policies/data-sharing-mining/>). Reference(s) to datasets should also be included in the reference list of the article with DOIs (where available).

If you wish to submit your data to Dryad (<http://datadryad.org/>) and have not already done so you can submit your data via this link [http://datadryad.org/submit?journalID=RSPB&manu=\(Document not available\)](http://datadryad.org/submit?journalID=RSPB&manu=(Document%20not%20available)), which will take you to your unique entry in the Dryad repository.

Online supplementary material will also carry the title and description provided during submission, so please ensure these are accurate and informative. Note that the Royal Society will not edit or typeset supplementary material and it will be hosted as provided. Please ensure that

the supplementary material includes the paper details (authors, title, journal name, article DOI). Your article DOI will be 10.1098/rspb.[paper ID in form xxxx.xxxx e.g. 10.1098/rspb.2016.0049].

Please submit a copy of your revised paper within three weeks. If we do not hear from you within this time your manuscript will be rejected. If you are unable to meet this deadline please let us know as soon as possible, as we may be able to grant a short extension.

Best wishes,
Dr Locke Rowe
mailto: proceedingsb@royalsociety.org

Associate Editor

Comments to Author:

This paper investigates the link between chemical cues and sociality in lizards and I really enjoyed reading this paper. While saying this I think it is important to make clearer in the introduction already that you used the existence of epidermal glands as a proxy for the investment in chemical signaling, but that other species also invest in chemical communication without investing in epidermal glands.

Reviewer(s)' Comments to Author:

Referee: 1

Comments to the Author(s)

I have now read the paper entitled 'Investment in chemical signalling facilitates the evolution of sociality in lizards' submitted by Baeckens and Whiting. I found the question interesting and the analyses and conclusions compelling. The analyses are appropriate and interpreted carefully. I only have minor comments for the authors as listed below.

line 36: of 910 lizard species

line 39: rate ... 15 time higher (rates are not likelihoods)

line 70: add a space before the reference number

lines 112-114: lizards also use olfaction, yet this remains dramatically understudied but should be mentioned. Especially in geckos this may be an important sense.

line 131: why use the Pyron et al phylogeny and not some of the more recent properly time-calibrated and more complete phylogenies that are out there?

line 146: briefly explain 'D' here as it is the first time it is mentioned.

line 193: findings of correlated evolution ...

line 204: throughout lizard evolutionary history.

line 207: I would delete 'favourable'

lines 239-242: yet scent marks offer less flexibility ... they stay longer and also provide excellent cues for predators like snakes. So they also have some disadvantages.

line 267: I was a little confused here ... you say 25% and list 4341 species ... is that squamates or lizards ? please clarify.

line 289: one another, and ...

line 299: study design [67,81]. This makes ...

figure 1: great figure ! the circles above the phylogeny are a little small for old people like me so me make them a tiny bit larger ?

Referee: 2

Comments to the Author(s)

This study (“Investment in chemical signalling facilitates the evolution of sociality in lizards”) tests the hypothesis that social grouping in lizards (defined as groups containing young and adults) is correlated with the presence of chemical signal-producing epidermal glands, which the authors use as a proxy to infer investment in chemical signalling. The authors used phylogenetic comparative methods to assess the correlation between social grouping and the presence of epidermal glands among 910 lizard species. The results from three different statistical approaches all indicate that social grouping is more likely to arise in lineages possessing epidermal glands, supporting their hypothesis. I enjoyed reviewing this manuscript and think that, pending major revision, it is potentially a good fit for publication in *Proceedings B*. However, I have major concerns that I feel need to be addressed. These concerns revolve around how the study is framed, how results are interpreted, and errors and other aspects of the underlying data. I also have a few questions about the statistics used, but am pleased to say the approach seems good to me overall, although I admit I am not a specialist in these specific analyses.

I contest the claim that the presence of epidermal glands reflects investment in chemical signalling, and thus think that the main conclusion of this paper is overstated (although I find the results themselves nevertheless interesting). While the presence/absence of glands might reflect chemical signal investment among some closely related species (e.g., *Liolaemus*; Ruiz-Monachesi et al. 2020, *Amphibia-Reptilia*), skinks, snakes, varanids, and other diverse squamate groups use chemical signals (including in social interactions) and yet do not possess epidermal glands. You acknowledge exactly these points in the Discussion, which I was very pleased to see, but a reader would not get any impression of these caveats reading your title, abstract, or introduction. I think the paper (including the title) needs to be edited so that these are not conflated. See line number comments below for some specific examples.

This is a correlational study, as are almost all macroevolutionary studies. You are usually careful to phrase things so as not to conflate correlation with causation, with the major exception of your paper’s title, and a few smaller instances (see line comments). You cannot be certain that the presence of epidermal glands is not correlated with another trait that is directly influencing the evolution of sociality. Perhaps sociality is actually less likely to evolve in shrub-dwelling lizards (e.g., *Anolis* and chameleons), which appear less likely to possess epidermal glands (Baeckens et al. 2015). This is just the first example that came to mind; I’m not suggesting that claim is likely true. Conflating correlation and causation is, unfortunately, extremely common among macroevolutionary papers, but is nevertheless worth avoiding.

For some species with both epidermal glands and social grouping, only males possess epidermal glands (e.g., *Naultinus*, *Hoplodactylus*, *Rhacodactylus*; I assume there are more), and evidence suggests that epidermal glands in lizards do not differentiate until the onset of sexual maturity (briefly reviewed in Mayerl et al. 2015, *Amphibia-Reptilia*). Assuming I’m not mistaken about these claims, how do you reconcile these observations with your results? This study specifically tests the correlation between epidermal glands and social grouping defined as groups containing adults and juveniles. How can epidermal glands mediate interactions among adults and juveniles

if juveniles do not possess functioning glands, and if (for some species) only a single sex possesses them? This is another reason why I urge caution with inferring causation, and I think these points are worth discussing.

While going through the data I found some errors. Both *Amblyrhynchus cristatus* and *Brachylophus* (2 spp.) are listed as not possessing epidermal glands, but they do, with papers published on the composition of secretions from these glands in *Amblyrhynchus* (<https://peerj.com/articles/3689/>). These species are early on alphabetically and so caught my eye easily, but it's worth checking the data more thoroughly. (ref for *Brachylophus vitiensis*: <https://journals.plos.org/plosone/article?id=10.1371/journal.pone.0073127>).

I looked at the data and references from Halliwell et al., with a focus on species I'm most familiar with, just to get a personal impression of how accurate the claims of social grouping are. In your study and in Halliwell et al. 2017, social grouping is defined as: "social groups containing both adults and juveniles" (quoted from Results section of Halliwell et al. 2017). Considering this, I take issue with the classification of *Underwoodisaurus milii* and *Christinus marmoratus* as species with social grouping. Their classification as such is based on Kearney et al. 2001 (*Herpetologica*), which states for *U. milii* (as *Nephrurus milii*): "For example, adult females of *N. milii* were rarely found with juvenile conspecifics", and, "...members of an aggregation tended to resemble each other in body size". While rarely doesn't mean never, this does not support that this species has inter-generational social grouping. In that same study, there was evidence of aggregations in *Christinus marmoratus*, with the main finding that groups tended to contain only a single male but otherwise reflected what would be expected by chance. I'm not contesting that these species aggregate, but there's not good evidence that there is social grouping as defined here. Considering that these species lack epidermal glands, changing their social grouping classification would make your results even stronger. But the other consequence of all this is that my confidence in the classifications made by Halliwell et al. is somewhat lower. Halliwell et al. classified many species based on a single book (Somma 2003), that for many species (e.g., *Gekko gecko*) cites mainly anecdotal observations in terraria. Because social grouping in lizards appears relatively rare, even a few misclassifications can bias the results of your statistical tests, and so it's worth worrying about and probably checking – and at the very least, acknowledging as a limitation.

Why were snakes excluded from your analyses? They were included in Halliwell et al. 2017, they demonstrate social grouping as it is treated herein, and they are simply a diverse radiation of reduced-limbed lizards. Other reduced-limbed radiations such as pygopodids and amphisbaenids are also included here, so I see no sound reason to exclude snakes from the analysis. That they do not possess epidermal glands would not be a good reason because other diverse clades lacking glands (and sociality; e.g., anoles & chameleons) are included.

According to Garcia-Roa et al. 2017, 25% of lizards have epidermal glands, but for the species included in your study, 31.7% have glands. Thus, your data are not a random sample of the phylogeny, and some groups are underrepresented (e.g., skinks). Could this sampling bias influence the results of your statistical analyses? I'm not familiar enough with the underlying assumptions of these analyses to make specific comments, but my impression is that – as with statistics generally – a non-random sample will yield spurious conclusions. If this is the case it should be discussed as a limitation of the study, and if it is not the case then I just need some reassurance. On a related note, there's contention about the validity of ancestral state reconstructions, with a recent study (Holland et al. 2020, *Scientific Reports*) indicating that they are often unreliable when traits are not selectively neutral. Notably, SSE models (including MuSSE, which was used here) are prone to high type I error rates when the trait does not influence speciation and/or extinction rates. How does MuSSE perform with your data compared to other models?

The caption for Figure 1 indicates that branches should be coloured according to the ancestral state reconstruction, but from what I can see all branches are black.

I would like to see all R code and a file containing the pruned phylogeny added to the electronic supplementary material for scrutiny and reproducibility.

LN27: I suggest striking this first sentence. Your study addresses the correlates of sociality, but not *why* organisms are social, and so it seems a little out of place — plus the next sentence works well as an opener in my opinion.

LN35: As per my earlier comment, you test the correlation with epidermal glands, not with investment in chemical signalling.

LN35–37: Here you say that 32% of 910 species have epidermal glands, and *of those*, 42% are social — but this directly contradicts your raw data. In fact, only 27 of the 288 gland-bearing species are considered social here, which is 9% of species with epidermal glands.

LN40: Conflating correlation with causation. Your results highlight the “potential” importance of chemical signalling via epidermal glands. I do share your hypothesis that chemical communication is likely very important for the evolution of sociality, but these results (if robust) are correlational.

LN60–61: I’m not a fan of this opening sentence. It comes off as a bit sensationalist, like the opening of a David Attenborough documentary. Also, to be a pedant, those organisms all have other things in common, too. This is just my superficial impression — I leave the writing style entirely to you.

LN62: Stray comma. I suggest the comma goes between “sociality” and “including”.

LN64: “Biologist” should be plural, “biologists”.

LN90: No need for the hyphen between “short-distances”.

LN92: You’re not really arguing this, rather, you’re proposing it as a hypothesis.

LN96: “...for, for...”

LN104–107: And yet agamids overwhelmingly possess epidermal glands, while skinks lack them entirely. This calls into question LN110–112, which posits that the presence of epidermal glands reflects investment in chemical signalling, even though you’ve just mentioned that skinks (which lack epidermal glands) are more chemically oriented. I realise you qualify this with the comment about needing more empirical support, but it seems clear that the presence/absence of epidermal glands is not a good indication of “investment” in chemical signalling. I think this is a misleading way to frame the problem, and leads to unfounded conclusions when interpreting your results. See my other comments on this matter.

LN113–114: There’s evidence suggesting that geckos are olfactory specialists that do not rely heavily on tongue-flicking (Schwenk 1993, *J. Zoology*), although I acknowledge that we need a lot more data on this.

LN126: Consider adding the skink *Nangura spinosa* (sometimes *Concinnia spinosa*) to your analysis. It is in the phylogeny of Pyron et al. and has social grouping, although this species wasn’t included in Halliwell et al., possibly because the references for it (see following ref) are mainly presented in government reports and field guides (e.g., Wilson & Swan 2017). Department of Environment and Resource Management 2010. Recovery plan for the Nangur spiny skink (*Nangura spinosa*). Report to the Department of Sustainability, Environment, Water, Population and Communities, Canberra. Department of Environment and Resource Management, Brisbane.

LN127: Should be "...phylogenetic comparative analysis..."

LN131: Needs a full stop in "al."

LN132-135: I think the definition of social grouping used herein should be explicitly stated in the Introduction, and maybe in the Abstract if there's room for it. Social grouping can be conceptualised in various ways, so I think it's best to be as up-front about it as possible to avoid confusion (and just make it easier for the reader).

LN139: A lot of the classifications regarding epidermal glands comes from Mayerl et al. 2017. Is the data from this paper publicly available? I can't find it.

LN141: Does "epidermal glands" include generation glands, or only epidermal follicular pores?

LN144: In the text file with your raw data, the species column states names are from Pyron et al. 2011, but I assume you actually mean Pyron et al. 2013 [ref 44]?

LN165: I assume you're saying you used a chi-squared test? If so I suggest rephrasing to make more obvious, e.g., "We compared the likelihoods of these two models using a chi-squared test..."

LN170: Can you explain this analysis better? I had to read Halliwell et al's methods to understand what was going on here (assuming you did this the same way they did).

LN171: I'm unfamiliar with the details of make.simmap, but is there a reason the permutations are written as 100 x 100 rather than 10,000?

LN181: "...with respect to the phylogeny."

LN197: I suggest striking "highly".

LN199: Degrees of freedom are not meaningful for permutational analyses. The p-value is calculated from the permutation distribution.

LN203: Conflating correlation with causation. Remove "facilitated" and rephrase.

LN203-204: Where has this pair-bonded and kin-based stuff come from? The social grouping classification is based on groups consisting of adults and juveniles as per Halliwell et al. 2017, not pair-bonded adults or anything to do with kin (except in some of their stable social grouping stuff, but that's not used here). This statement is beyond the scope of this study. Same comment goes for LN213-215. For many of the studies Halliwell et al. used to classify social grouping, there was actually no way to know whether the observed juveniles were actually the offspring of the adults they were found with, or kin.

LN212-225: What about the fact that for many species only males possess epidermal glands, and that juveniles of lizards possessing epidermal glands will not have fully formed and functional epidermal glands? How can epidermal glands then mediate social interactions in these cases?

LN209: It highlights the *potential* role.

LN226: Must communication really be continuous?

LN234: Regarding "juveniles and adults" — the cited study looks at female discrimination of "young" and "old" males, but there is no indication that these are sexually immature juveniles. In fact, the words "juvenile" and "immature" do not appear in the manuscript (except in the reference list).

LN262–266: These references do not demonstrate that epidermal glands are *the* main source of chemical signals in lizards, even if that claim is frequently made within them. Three of these are reviews (which of course do review how epidermal glands are an *important* source of chemical signals), and one is a macroevolutionary study (although Mayerl et al. also has a bit of macro in it). None of these references test that epidermal glands are behaviourally more important than other sources of chemical signals (e.g., urodaeal glands, faeces, etc.). If there are studies that support this claim broadly across lizards please cite them directly. I can't help but get the impression that ref 68 is a gratuitous self-citation here.

LN270–272: This is false. We know that skinks, varanids, and snakes rely heavily on chemical signalling, and none of these possess epidermal glands. It might be true in specific circumstances (e.g., Ruiz-Monachesi et al. 2020), but is demonstrably false in broad terms. And you go on over the rest of the paragraph to acknowledge this — there are many sources of import, socially mediating chemical signals in lizards. Thus, is there any value in stating something like: “Because of their significance in chemical signalling, the absence of epidermal glands in lizards likewise suggests less reliance on chemical signals as an information source” before diving in to these caveats?

LN279: I suggest replacing “depauperated” with a more commonly used synonym, e.g., “reduced”.

LN294–305: You say tongue-flicking is more suitable in one sentence, then in the next sentence contradict yourself and say it's too biased and flawed. You could simply say that baseline tongue-flick rates are an alternative proxy (without saying it's probably better), and then outline its strengths and weaknesses without contradicting yourself.

LN305–309: You do not show any of this. You find a correlation between the presence of epidermal glands and the evolution of social grouping defined as associations between adults and juveniles. This is not evidence that epidermal glands facilitated the evolution of social grouping behaviour (conflating correlation and causation), and social grouping was not defined by pair-bonds or family units. Halliwell et al. give an additional and stricter classification of “stable social grouping”, but that is not what was analysed here.

LN327: Branch colours are not visible on the figure. All branches are black.

Referee: 3

Comments to the Author(s)

The authors are suggesting that the mode of communication facilitates the occurrence of sociality, i.e. that the initial presence of glands allows and contributes to the development of sociality. However, in principle, one would expect that signal complexity increases as the need for communication increases. Or more likely that there is a coevolution process in which an initial low investment in communication increases to a higher level of investment as sociality develops. I was not convinced that these alternative scenarios were properly refuted or even discussed.

Minor comments

Abstract: I find this sentence in the Abstract confusing: “We found that roughly 32% of 910 species invest in signal-producing epidermal glands and of these, approximately 42% form social groups.” This seems to imply that 58% of species with epidermal glands, i.e. a majority of them, do not form social groups, and this seems to contradict your main conclusion. People reading the abstract without having read the rest of the paper will get confused.

Lines 81-82: "Whether the mode of communication has influenced social evolution in terrestrial vertebrates remains largely unexplored." No references are given here, which overemphasizes the "unexplored" claim to a point that is not correct. For example, there are studies in strepsirrhine primates on the matter.

Line 96: correct "for, for"

Author's Response to Decision Letter for (RSPB-2020-2438.R0)

See Appendix A.

RSPB-2020-2438.R1 (Revision)

Review form: Reviewer 2

Recommendation

Major revision is needed (please make suggestions in comments)

Scientific importance: Is the manuscript an original and important contribution to its field?

Good

General interest: Is the paper of sufficient general interest?

Good

Quality of the paper: Is the overall quality of the paper suitable?

Good

Is the length of the paper justified?

Yes

Should the paper be seen by a specialist statistical reviewer?

No

Do you have any concerns about statistical analyses in this paper? If so, please specify them explicitly in your report.

No

It is a condition of publication that authors make their supporting data, code and materials available - either as supplementary material or hosted in an external repository. Please rate, if applicable, the supporting data on the following criteria.

Is it accessible?

Yes

Is it clear?

Yes

Is it adequate?

Yes

Do you have any ethical concerns with this paper?

No

Comments to the Author

I'm pleased to see that most of my comments have been addressed, and the authors have clarified a few points, including some that I was confused or mistaken about. However, I disagree with a few of the authors' responses, and still have two major comments, along with a few minor ones. Of these comments, I feel that the first, concerning the exclusion of snakes, is the main issue that should be addressed. I think doing so will improve the generalisability of the study, and I would be pleased to see it subsequently published.

My other comments largely concern recognition and discussion of limitations. I know I'm very critical in these, but I want to stress that I don't think these limitations make the study unpublishable, nor do they need to be explained away. I simply think they are worth openly recognising and discussing because they highlight the knowledge gaps that are crucial for future research to address.

Major comments

My first major comment has to do with the exclusion of snakes. You provided the following response to my question about why snakes were excluded:

"Because of the single gland loss (and no re-emergence) and because we felt we couldn't provide enough support for the idea of epidermal glands as a proxy for chemical signal investment in snakes, we decided to focus solely on lizards in this study. Note, however, that we only interpret our data and limited our conclusions to lizards, and that we do not make any claims about the relationship between snake sociality and chemical abilities."

It strikes me that this exact same line of reasoning could also apply to skinks: there are ~1,700 recognised species (16% of all squamates); no known species possess generation or follicular epidermal glands (single loss, no re-emergences); many studies demonstrating the role of scent in skink social interactions; and perhaps the most incredible examples of complex sociality in squamates (i.e., *Egernia* group skinks). One could then argue the same for less diverse groups, such as varanids. Considering these points, I don't think your reasoning for excluding snakes is valid. While you say you limit interpretation of your results to lizards only, the fact is that snakes are lizards. By excluding them, you limit the generalisability of your results, thus limiting what is the most valuable aspect of macroevolutionary studies. It is clear that the presence of epidermal glands is not an accurate proxy for chemical signalling investment; this study can only make conclusions about the association between epidermal glands and social grouping. That remains the case whether or not snakes are included. I think by including snakes you can make your results more generalizable — and I would argue, more accurate — regardless of what those new results may be.

My second major comment is in regards to the social grouping data and its classification. As I stated in the previous review, I am uneasy about the overall lack of rigour stemming from the use of the data from Halliwell et al. I acknowledge that the available information is extremely limited; accordingly, Halliwell et al. were necessarily not very strict in what warranted evidence of social grouping (see 'Assigning social grouping' section of their Methods). Furthermore, scrutiny of sources was relaxed because much of the literature was unavailable: from the methods of Halliwell et al.: "...[We]... relied on the interpretation of authors when we could not access primary sources". I am uneasy when data are inscrutable, and using such data can perpetuate errors from publication to publication. But I also recognise that one must do what they can with what's available. As such, I'd like to see more discussion of the limitations stemming from all this.

This brings me to a related point. The underlying data reflect a whole assortment of sources, some of which are detailed studies, but many of which are anecdotal observation from the field or of captive animals. Detailed studies are obviously more reliable than anecdote. For example, if any single observation of social tolerance warrants being classified as social grouping, *Underwoodisaurus milii* and *Christinus marmoratus* would indeed be classified as such (contrary to my comments in the last review and your subsequent edits). But more detailed evidence suggests that, at least in *U. milii*, aggregations of young and adults are generally avoided – but such aggregations are still sometimes observed! Thus, there is a mismatch between social classification based on anecdote, and social classification based on more rigorously collected data. My concern is that many species are misclassified based on anecdote. I admit that I'm not sure what practical steps you can take to address this here, other than perhaps reviewing the literature yourself and pruning those species without sufficient evidence, which might leave with very little left to analyse.

Minor comments

From your response: "Yet, 'facilitates' does not imply causality, rather it indicates that the emergence of sociality throughout the evolutionary history of lizards occurs at a higher rate in lineages carrying epidermal glands relative to those without epidermal glands"

'Facilitates' does in fact imply a degree of causality. Facilitate is defined as a verb that "makes an action or process easier." To make something easier, it must influence (causally) the outcome of the action or process. You could say, however, that the association between glands and grouping suggests that glands might facilitate grouping. I simply suggest qualifying statements such as the one on line 362 to better reflect the uncertainty surrounding correlational analyses. As I stated in the last review, in most cases you are careful not to conflate correlation with causation; I am just being nit-picky about the few exceptions.

Regarding the ancestral state reconstructions via stochastic character mapping, you respond:

"We rather disagree that showing the degrees of freedom is not meaningful, as we believe it informs the reader on the number of permutations used. We prefer to include the degrees of freedom, unless the reviewer and editor strongly advice not too."

Ok I think I was confused here. I was previously under the false impression that this was a permutation test (in which there are no 'degrees of freedom' in the sense of null-hypothesis significance testing), but I can see now that's not the case. Although, I admit I'm still a bit confused about this analysis even with the additional details you've given. Specifically, how is the hypothesis test being done? That you report a t-score suggests a t-test. You state: "The similarity of reconstructions between map sets was estimated by computing the mean and distribution of overlap in stochastic maps." My impression isn't that you tested for a difference between the map sets, but rather that you tested for a difference between the distribution of overlap in the map sets against an expectation of 0.5 (is this correct?). Can you add detail explicitly stating how the hypothesis test was done? I hope I'm not just misinterpreting everything; and I apologise if I am.

The following is your response to my question about species with social grouping where the female lacks glands:

"The reviewer is correct in that there are species known for which only one of the sexes carries active epidermal glands (Chuahhan 1986; Dujsebayaeva et al 2009; Lobo et al 2012). However, the large majority of studies simply find sexual differences in the anatomy (size) of epidermal pores, but whether this is merely a body size-effect and, more importantly, whether these pore size differences impact the secretion production, "pheromonal activity", or signalling potential of the glands is only known for a small number of species with mixed results (Cole 1966; Alberts et al 1992, 1993; Van Wyk & Mouton 1992; Jared et al 1999; Khannoon et al 2013; Imparato et al 2007;

Baeckens 2017). To show that we acknowledge that in some species there are sexual differences in gland activity, we wrote in the discussion L322: “Other work suggests that some species rely on a combination of chemicals from different sources to enhance signal effectiveness or to broadcast different messages at once [40,85]. Skin extracts and epidermal gland secretions of male *Liolaemus tenuis* lizards, for instance, are found to mainly trigger marking behaviour in female conspecifics, whilst the scent of their faeces elicited predominantly display behaviour [40]. Exploiting multiple chemical sources may be particularly useful in lizard species for which active epidermal glands are restricted to a single sex only. In species where females lack active epidermal glands, such as *Podarcis hispanicus*, male conspecific can extract information from female cloacal and body odours to determine, for instance, reproductive condition [86], while females can profit from the information-rich epidermal gland secretions to assess, for example, a male’s immune response [87].”

Your response here hasn’t addressed my concern: How do you reconcile your interpretation that glands might better facilitate social grouping with the observation that, in some of these species, females lack glands? I acknowledge that in many species the glands of females are smaller rather than entirely absent — but the exceptions matter if they are also social species. Of the 27 species with glands and social grouping, how many species have females lacking these glands? If it’s a very small number, then it’s probably still consistent with your results; but if it’s closer to half then it seems equivocal. (Off the top of my head I can see ~10 out of 27 spp. for which I think females lack functioning glands, but I could very well be wrong.) If glands do better facilitate the evolution of social grouping and mediate social tolerance among adults and young, then we’d expect both sexes to possess them. You go on to explain how there are potentially other sources of chemical signals that could mediate interactions, but this is only tangentially relevant to my comment, which specifically concerns your results for epidermal glands, and not chemical signalling investment more broadly. Ok, I realise this is getting long for a ‘minor comment’... I recommend that, instead of trying to explain away this (potential) inconsistency by invoking other chemical sources to fit it into your hypothesis, you simply 1) state that glands can be absent in females, and state the number of species with social grouping in which this is the case; 2) discuss whether this number is small or large, and thus how consistent it is with your hypothesis; and 3) if it is inconsistent, highlight how interesting it is, and what it might mean for future research that tries to better understand chemically-mediated social tolerance and sociality.

Line comments

LN113: I still contest the claim that epidermal glands are the “main source” — again, skinks and snakes (which are lizards, regardless of whether they’ve been included herein) demonstrate this. More accurately, you could say that glands are likely the main source in species that possess these glands, which is what the study you summarise on *Liolaemus tenuis* actually supports. As far as I’m aware, the claim has never been tested among squamates more broadly.

LN128–129: Would you consider social grouping as defined here to be “complex sociality”? To me, it seems to be close to the simplest end of the simple–complex spectrum of sociality.

LN213–214: Are there confidence intervals for these rate estimates? I assume there must be some degree of uncertainty as a result of uncertainty in the ancestral state reconstructions.

LN230: I suggest either replacing i.e. with e.g., or else entirely removing “kin-based”. For many species classified as social grouping, there was no information whether grouping was between kin. It seems more likely that’s the case, but it nevertheless remains an untested assumption for many species.

LN245: Suggest rephrasing: “...with their mothers, and potentially also their fathers in pair-bonded species.” Or something along those lines — the previous wording was confusing when I first read it.

LN309: Full-stop missing at end of sentence (after the citations).

LN309 & 313: Egernia should be italicised.

LN287–366: This is a very long paragraph. I suggest breaking it up into 2–3 paragraphs if it's logical to do so.

Decision letter (RSPB-2020-2438.R1)

19-Jan-2021

Dear Dr Baeckens

I am pleased to inform you that your Review manuscript RSPB-2020-2438.R1 entitled "Investment in chemical signalling glands facilitates the evolution of sociality in lizards" has been accepted for publication in Proceedings B.

The referee(s) do not recommend any further changes. Therefore, please proof-read your manuscript carefully and upload your final files for publication. Because the schedule for publication is very tight, it is a condition of publication that you submit the revised version of your manuscript within 7 days. If you do not think you will be able to meet this date please let me know immediately.

To upload your manuscript, log into <http://mc.manuscriptcentral.com/prsb> and enter your Author Centre, where you will find your manuscript title listed under "Manuscripts with Decisions." Under "Actions," click on "Create a Revision." Your manuscript number has been appended to denote a revision.

You will be unable to make your revisions on the originally submitted version of the manuscript. Instead, upload a new version through your Author Centre.

1) A text file of the manuscript (doc, txt, rtf or tex), including the references, tables (including captions) and figure captions. Please remove any tracked changes from the text before submission. PDF files are not an accepted format for the "Main Document".

2) A separate electronic file of each figure (tiff, EPS or print-quality PDF preferred). The format should be produced directly from original creation package, or original software format. Please note that PowerPoint files are not accepted.

3) Electronic supplementary material: this should be contained in a separate file from the main text and the file name should contain the author's name and journal name, e.g. `authorname_procb_ESM_figures.pdf`

All supplementary materials accompanying an accepted article will be treated as in their final form. They will be published alongside the paper on the journal website and posted on the online figshare repository. Files on figshare will be made available approximately one week before the accompanying article so that the supplementary material can be attributed a unique DOI. Please see: <https://royalsociety.org/journals/authors/author-guidelines/>

4) Data-Sharing and data citation

It is a condition of publication that data supporting your paper are made available. Data should be made available either in the electronic supplementary material or through an appropriate

repository. Details of how to access data should be included in your paper. Please see <https://royalsociety.org/journals/ethics-policies/data-sharing-mining/> for more details.

If you wish to submit your data to Dryad (<http://datadryad.org/>) and have not already done so you can submit your data via this link <http://datadryad.org/submit?journalID=RSPB&manu=RSPB-2020-2438.R1> which will take you to your unique entry in the Dryad repository.

Once again, thank you for submitting your manuscript to Proceedings B and I look forward to receiving your final version. If you have any questions at all, please do not hesitate to get in touch.

Sincerely,

Dr Locke Rowe
Editor, Proceedings B
<mailto:proceedingsb@royalsociety.org>

Associate Editor Board Member: 1

Comments to Author:

The authors did a good job and the manuscript improved through the revision process. I agree with the referee that including snakes would make the study much stronger, and therefore I think the authors should consider to include snakes. However, I see the argument of the authors and I think the exclusion of snakes is not fatal.

Nevertheless, the authors should address the other comments of the referee and add some sentences about the limitations of the data.

Reviewer(s)' Comments to Author:

Referee: 2

Comments to the Author(s)

I'm pleased to see that most of my comments have been addressed, and the authors have clarified a few points, including some that I was confused or mistaken about. However, I disagree with a few of the authors' responses, and still have two major comments, along with a few minor ones. Of these comments, I feel that the first, concerning the exclusion of snakes, is the main issue that should be addressed. I think doing so will improve the generalisability of the study, and I would be pleased to see it subsequently published.

My other comments largely concern recognition and discussion of limitations. I know I'm very critical in these, but I want to stress that I don't think these limitations make the study unpublishable, nor do they need to be explained away. I simply think they are worth openly recognising and discussing because they highlight the knowledge gaps that are crucial for future research to address.

Major comments

My first major comment has to do with the exclusion of snakes. You provided the following response to my question about why snakes were excluded:

“Because of the single gland loss (and no re-emergence) and because we felt we couldn’t provide enough support for the idea of epidermal glands as a proxy for chemical signal investment in snakes, we decided to focus solely on lizards in this study. Note, however, that we only interpret our data and limited our conclusions to lizards, and that we do not make any claims about the relationship between snake sociality and chemical abilities.”

It strikes me that this exact same line of reasoning could also apply to skinks: there are ~1,700 recognised species (16% of all squamates); no known species possess generation or follicular epidermal glands (single loss, no re-emergences); many studies demonstrating the role of scent in skink social interactions; and perhaps the most incredible examples of complex sociality in squamates (i.e., *Egernia* group skinks). One could then argue the same for less diverse groups, such as varanids. Considering these points, I don’t think your reasoning for excluding snakes is valid. While you say you limit interpretation of your results to lizards only, the fact is that snakes are lizards. By excluding them, you limit the generalisability of your results, thus limiting what is the most valuable aspect of macroevolutionary studies. It is clear that the presence of epidermal glands is not an accurate proxy for chemical signalling investment; this study can only make conclusions about the association between epidermal glands and social grouping. That remains the case whether or not snakes are included. I think by including snakes you can make your results more generalizable – and I would argue, more accurate – regardless of what those new results may be.

My second major comment is in regards to the social grouping data and its classification. As I stated in the previous review, I am uneasy about the overall lack of rigour stemming from the use of the data from Halliwell et al. I acknowledge that the available information is extremely limited; accordingly, Halliwell et al. were necessarily not very strict in what warranted evidence of social grouping (see ‘Assigning social grouping’ section of their Methods). Furthermore, scrutiny of sources was relaxed because much of the literature was unavailable: from the methods of Halliwell et al.: “...[We]... relied on the interpretation of authors when we could not access primary sources”. I am uneasy when data are inscrutable, and using such data can perpetuate errors from publication to publication. But I also recognise that one must do what they can with what’s available. As such, I’d like to see more discussion of the limitations stemming from all this.

This brings me to a related point. The underlying data reflect a whole assortment of sources, some of which are detailed studies, but many of which are anecdotal observation from the field or of captive animals. Detailed studies are obviously more reliable than anecdote. For example, if any single observation of social tolerance warrants being classified as social grouping, *Underwoodisaurus milii* and *Christinus marmoratus* would indeed be classified as such (contrary to my comments in the last review and your subsequent edits). But more detailed evidence suggests that, at least in *U. milii*, aggregations of young and adults are generally avoided – but such aggregations are still sometimes observed! Thus, there is a mismatch between social classification based on anecdote, and social classification based on more rigorously collected data. My concern is that many species are misclassified based on anecdote. I admit that I’m not sure what practical steps you can take to address this here, other than perhaps reviewing the literature yourself and pruning those species without sufficient evidence, which might leave with very little left to analyse.

Minor comments

From your response: “Yet, ‘facilitates’ does not imply causality, rather it indicates that the emergence of sociality throughout the evolutionary history of lizards occurs at a higher rate in lineages carrying epidermal glands relative to those without epidermal glands”

'Facilitates' does in fact imply a degree of causality. Facilitate is defined as a verb that "makes an action or process easier." To make something easier, it must influence (causally) the outcome of the action or process. You could say, however, that the association between glands and grouping suggests that glands might facilitate grouping. I simply suggest qualifying statements such as the one on line 362 to better reflect the uncertainty surrounding correlational analyses. As I stated in the last review, in most cases you are careful not to conflate correlation with causation; I am just being nit-picky about the few exceptions.

Regarding the ancestral state reconstructions via stochastic character mapping, you respond: "We rather disagree that showing the degrees of freedom is not meaningful, as we believe it informs the reader on the number of permutations used. We prefer to include the degrees of freedom, unless the reviewer and editor strongly advice not too."

Ok I think I was confused here. I was previously under the false impression that this was a permutation test (in which there are no 'degrees of freedom' in the sense of null-hypothesis significance testing), but I can see now that's not the case. Although, I admit I'm still a bit confused about this analysis even with the additional details you've given. Specifically, how is the hypothesis test being done? That you report a t-score suggests a t-test. You state: "The similarity of reconstructions between map sets was estimated by computing the mean and distribution of overlap in stochastic maps." My impression isn't that you tested for a difference between the map sets, but rather that you tested for a difference between the distribution of overlap in the map sets against an expectation of 0.5 (is this correct?). Can you add detail explicitly stating how the hypothesis test was done? I hope I'm not just misinterpreting everything; and I apologise if I am.

The following is your response to my question about species with social grouping where the female lacks glands:

"The reviewer is correct in that there are species known for which only one of the sexes carries active epidermal glands (Chuahhan 1986; Dujsebayaeva et al 2009; Lobo et al 2012). However, the large majority of studies simply find sexual differences in the anatomy (size) of epidermal pores, but whether this is merely a body size-effect and, more importantly, whether these pore size differences impact the secretion production, "pheromonal activity", or signalling potential of the glands is only known for a small number of species with mixed results (Cole 1966; Alberts et al 1992, 1993; Van Wyk & Mouton 1992; Jared et al 1999; Khan Noon et al 2013; Imparato et al 2007; Baeckens 2017). To show that we acknowledge that in some species there are sexual differences in gland activity, we wrote in the discussion L322: "Other work suggests that some species rely on a combination of chemicals from different sources to enhance signal effectiveness or to broadcast different messages at once [40,85]. Skin extracts and epidermal gland secretions of male *Liolaemus tenuis* lizards, for instance, are found to mainly trigger marking behaviour in female conspecifics, whilst the scent of their faeces elicited predominantly display behaviour [40]. Exploiting multiple chemical sources may be particularly useful in lizard species for which active epidermal glands are restricted to a single sex only. In species where females lack active epidermal glands, such as *Podarcis hispanicus*, male conspecific can extract information from female cloacal and body odours to determine, for instance, reproductive condition [86], while females can profit from the information-rich epidermal gland secretions to assess, for example, a male's immune response [87]."

Your response here hasn't addressed my concern: How do you reconcile your interpretation that glands might better facilitate social grouping with the observation that, in some of these species, females lack glands? I acknowledge that in many species the glands of females are smaller rather than entirely absent – but the exceptions matter if they are also social species. Of the 27 species with glands and social grouping, how many species have females lacking these glands? If it's a very small number, then it's probably still consistent with your results; but if it's closer to half then it seems equivocal. (Off the top of my head I can see ~10 out of 27 spp. for which I think females lack functioning glands, but I could very well be wrong.) If glands do better facilitate the

evolution of social grouping and mediate social tolerance among adults and young, then we'd expect both sexes to possess them. You go on to explain how there are potentially other sources of chemical signals that could mediate interactions, but this is only tangentially relevant to my comment, which specifically concerns your results for epidermal glands, and not chemical signalling investment more broadly. Ok, I realise this is getting long for a 'minor comment'... I recommend that, instead of trying to explain away this (potential) inconsistency by invoking other chemical sources to fit it into your hypothesis, you simply 1) state that glands can be absent in females, and state the number of species with social grouping in which this is the case; 2) discuss whether this number is small or large, and thus how consistent it is with your hypothesis; and 3) if it is inconsistent, highlight how interesting it is, and what it might mean for future research that tries to better understand chemically-mediated social tolerance and sociality.

Line comments

LN113: I still contest the claim that epidermal glands are the "main source" – again, skinks and snakes (which are lizards, regardless of whether they've been included herein) demonstrate this. More accurately, you could say that glands are likely the main source in species that possess these glands, which is what the study you summarise on *Liolaemus tenuis* actually supports. As far as I'm aware, the claim has never been tested among squamates more broadly.

LN128–129: Would you consider social grouping as defined here to be "complex sociality"? To me, it seems to be close to the simplest end of the simple–complex spectrum of sociality.

LN213–214: Are there confidence intervals for these rate estimates? I assume there must be some degree of uncertainty as a result of uncertainty in the ancestral state reconstructions.

LN230: I suggest either replacing i.e. with e.g., or else entirely removing "kin-based". For many species classified as social grouping, there was no information whether grouping was between kin. It seems more likely that's the case, but it nevertheless remains an untested assumption for many species.

LN245: Suggest rephrasing: "...with their mothers, and potentially also their fathers in pair-bonded species." Or something along those lines – the previous wording was confusing when I first read it.

LN309: Full-stop missing at end of sentence (after the citations).

LN309 & 313: *Egernia* should be italicised.

LN287–366: This is a very long paragraph. I suggest breaking it up into 2–3 paragraphs if it's logical to do so.

Author's Response to Decision Letter for (RSPB-2020-2438.R1)

See Appendix B.

Decision letter (RSPB-2020-2438.R2)

21-Jan-2021

Dear Dr Baeckens

I am pleased to inform you that your manuscript entitled "Investment in chemical signalling glands facilitates the evolution of sociality in lizards" has been accepted for publication in Proceedings B.

Your article has been estimated as being 9 pages long. Our Production Office will be able to confirm the exact length at proof stage.

Open Access

Paper charges

Sincerely,

Proceedings B

Appendix A

Dear Dr. Rowe,

On behalf of my co-author and I, thank you very much for the opportunity to revise our manuscript, entitled “**Investment in chemical signalling glands facilitates the evolution of sociality in lizards**”. We have carefully read the insightful comments from the three reviewers and we have done our best to amend the text in the ways they suggest. In particular, we thank Referee 2 for the particularly thorough and constructive feedback, which we believe has greatly improved the quality of our manuscript. We also appreciated that all reviewers acknowledge the significance of our data and its importance to the field of evolutionary and behavioural ecology. While all reviewers found significant merit in our manuscript, they pointed out several issues that needed to be addressed before they could recommend its publication. Specifically, based on the reviewers’ comments, we paid particular attention to two key points (1) “the use (and drawbacks) of epidermal glands as a proxy of investment in chemical signalling should be mentioned earlier in the manuscript” and (2) “be careful with conflating correlation and causation”, which we have addressed in the revised manuscript and below.

Again, we appreciate the reviewers’ helpful comments and would be happy to respond to further suggestions they may have for improving the manuscript.

Sincerely,

Simon Baeckens

AUTHORS' RESPONSE TO THE REVIEWERS' COMMENTS

ASSOCIATE EDITOR

General comment:

(#1) This paper investigates the link between chemical cues and sociality in lizards and I really enjoyed reading this paper. While saying this I think it is important to make clearer in the introduction already that you used the existence of epidermal glands as a proxy for the investment in chemical signalling, but that other species also invest in chemical communication without investing in epidermal glands.

RESPONSE: First of all, we are delighted that the associate editor really enjoyed reading our work. Secondly, we understand the associate editor's concern about how lizards use chemical signalling more broadly and the need to make clear early in the introduction that we have used epidermal glands as our proxy for investment in signalling. Specifically, we made a minor change to the title to reflect this and we added a sentence to the abstract that we use the presence of epidermal glands as a proxy for chemical signalling investment. The title is changed to "*Investment in chemical signalling glands facilitates the evolution of sociality in lizards*". In the abstract, on line 35-39, we added "*We used the presence of epidermal glands as a proxy of chemical investment and considered social grouping as the occurrence of social groups containing both adults and juveniles. Based on a dataset of 911 lizard species, our models strongly supported correlated evolution between social grouping and chemical signalling glands.*" In addition, we clarified in the introduction (what we already comprehensively explain in the discussion) that there are also alternative sources of semiochemicals in lizards. On L109-117, we added the following: "*In the case of lizards that use chemical communication, several possible semiochemical sources may be responsible for signal production. While there is some evidence that cloacal exudates and faecal pellets may contain socially relevant information, the skin and the (generation and follicular) epidermal glands are considered the main sources of chemical signals in lizard communication, enabling mate assessment, individual recognition, species recognition, and sex identification (reviewed in [35–39]). Liolaemus tenuis females, for instance, are more attracted to substrates covered with male epidermal gland secretions than to substrates scent-marked with male skin extracts [40].*" With these changes, we believe we have made it quite clear, at the onset of the manuscript (and not only in the discussion), how we gauge chemical signalling investment.

General comment:

(#2) I have now read the paper entitled 'Investment in chemical signalling facilitates the evolution of sociality in lizards' submitted by Baeckens and Whiting. I found the question interesting and the analyses and conclusions compelling. The analyses are appropriate and interpreted carefully. I only have minor comments for the authors as listed below.

RESPONSE: Thank you for the positive feedback!

Minor comments:

(#3) line 36: of 910 lizard species

RESPONSE: As suggested, we added “lizard” before “species” in the abstract text.

(#4) line 39: rate ... 15 time higher (rates are not likelihoods)

RESPONSE: As suggested, we interchanged “more likely” by “higher”. Similar changes were made in the first paragraph of the discussion.

(#5) line 70: add a space before the reference number

RESPONSE: Done.

(#6) lines 112-114: lizards also use olfaction, yet this remains dramatically understudied but should be mentioned. Especially in geckos this may be an important sense.

RESPONSE: We agree with the reviewer’s comment and changed the sentence to the following: “Lizards sample substrate-bound or airborne chemicals in the environment using tongue-flicking (vomeroolfaction), or receive chemical-laden air through the nasal nares (olfaction s.s.) [43,44].” We also added an extra references, i.e. Schwenk 1993 (J Zool) on “Are geckos olfactory specialists?”, to back-up this statement.

(#7) line 131: why use the Pyron et al phylogeny and not some of the more recent properly time-calibrated and more complete phylogenies that are out there?

RESPONSE: Because the statistical approach to study social evolution used in this study largely follows that of Halliwell et al. (2017 Nat Comm), we decided also to use the same phylogenetic tree that Halliwell et al. used (i.e. Pyron et al. 2013). By doing so, conclusions made on lizard social evolution can be more easily compared between studies; the use of

the same phylogenetic tree facilitates interstudy comparisons. Pyron et al. (2013) generated their tree using a sequence dataset comprising of 12 genes for 4161 species, which is acknowledged as a comprehensive molecular phylogenetic estimate of lizard relationships, and therefore frequently used in phylogenetic comparative lizard studies, e.g. Stuart-Fox et al (Biol Rev 2020), Bar et al (Biol Rev 2020), Ramm et al (Biol Lett 2020), and Allen et al (Evolution 2020) to name a few recent ones. More recent efforts, such as the squamate phylogeny proposed by Zheng & Wiens (Mol Phyl Evol 2016), which is comprised of 52 genes and 4162 species, do not show drastic topological differences with the Pyron phylogeny. In fact, the tree by Pyron et al (2013) contains a larger proportion of monophyletic genera than the Zheng & Wiens tree—arguably because of the high proportion of missing molecular data in the Zheng & Wiens tree. Importantly, though, while the phylogenetic relationships among lizard species are generally well-supported and do not fundamentally vary between the squamate trees build by Pyron et al (2013) and Zheng & Wiens (2016), the relationships among snake species appear to be more equivocal. Some of the snake relationships as proposed by Pyron et al. (2013) have been challenged by Zheng & Wiens (2016), which then again have been questioned by Tonini et al (Cons Biol 2016) and Miralles et al (J Evol Biol 2018). Our study, however, focuses solely on lizards.

Nevertheless, to validate the use of the Pyron tree, we assessed the potential effect of the phylogenetic tree of choice on our comparative analyses and re-analysed our data using the Zheng & Wiens (2016) tree. Similar to the results obtained using the Pyron tree, Markov models supported the scenario of correlated evolution between social grouping and epidermal glands, in which the evolution of social grouping depends on the presence of epidermal glands (Table R1 below). Also the results of the MuSSE model acquired using the Zheng & Wiens tree showed to be highly significant ($\Delta AIC = 56.33$, $\chi^2 = 58.34$, $P < 0.001$) similar to the models using the Pyron tree ($\Delta AIC = 8.8$, $\chi^2 = 10.77$, $P = 0.001$; see manuscript). We found the same pattern for the third approach, which revealed a similarity of 0.62 ($t = 1454.5$, $P < 0.001$; using Zheng & Wiens 2016) between stochastic character map sets based on separate ancestral character state reconstructions of social grouping and epidermal glands. This result is highly similar to the Pyron tree analyses, which also revealed a mean similarity of 0.63 ($t = 1517.6$, $P < 0.001$; see manuscript). We can conclude that the use of different phylogenetic estimates (Pyron et al vs. Zheng & Wiens) did not affect or change the general co-evolutionary pattern between the two traits of study.

Table R1 — Performance of Markov models using the phylogenetic tree proposed by Zheng & Wiens (2016), showing similar results as with analyses using the Pyron tree (see manuscript).

Model		LogL	AIC	LL-ratio	P-value
Independent evolution		-251.817	511.635		
Dependent evolution	Correlated evolution	-246.570	509.149	10.495	0.033
	Change in SG depends upon state of EG	-248.353	508.707	6.928	0.031
	Change in EG depends upon state of SG	-251.070	514.138	1.497	0.473

(#8) line 146: briefly explain 'D' here as it is the first time it is mentioned.

RESPONSE: As suggested, we added some clarification and changed the sentence to: “Next, we tested for phylogenetic signal in the two variables ‘social grouping’ and ‘epidermal glands’ by calculating *D* (a measure of phylogenetic signal in a binary trait [46]) using the function ‘*phylo.d*’ (1000 permutations) in the *caper* package [47].”

(#9) line 193: findings of correlated evolution ...

RESPONSE: As suggested, we changed “*correlative*” to “*correlated*”.

(#10) line 204: throughout lizard evolutionary history.

RESPONSE: As suggested, we changed “*through*” to “*throughout*”.

(#11) line 207: I would delete 'favourable'

RESPONSE: As suggested, we deleted “*favourable*” from the sentence.

(#12) lines 239-242: yet scent marks offer less flexibility ... they stay longer and also provide excellent cues for predators like snakes. So they also have some disadvantages.

RESPONSE: Very true! Note that on line 265 we do write that scent-marks persist for long periods of time in the environment. On line 268, we then write that this can increase the probability of detection and accurate processing by conspecifics. The reviewer is correct that this also applies to detection by heterospecifics, including predators. Hence, we included this in the sentence, with the appropriate references on chemical eavesdropping. The sentence now reads: “*Scent-marks effectively transmit the same information both passively and continuously, thereby increasing the probability of detection and accurate processing by*

conspecifics [18,22] (but also heterospecifics [64,65]) and helps stabilise social systems [66–68].”

(#13) line 267: I was a little confused here ... you say 25% and list 4341 species ... is that squamates or lizards ? please clarify.

RESPONSE: As suggested, we now have clarified that the sample size refers to the study by García-Roa et al (2017), and that it refers to the number of lizard species. The sentence now reads: *“Based on the most comprehensive literature search to date (n = 4,341 lizard species), approximately 25% of all lizard species are equipped with follicular epidermal glands [39].”*

(#14) line 289: one another, and ...

RESPONSE: As suggested, we included a “,” between *“one another”* and *“and”*.

(#15) line 299: study design [67,81]. This makes ...

RESPONSE: As suggested, we removed *“which”* and started a new sentence.

(#16) figure 1: great figure ! the circles above the phylogeny are a little small for old people like me so me make them a tiny bit larger ?

“RESPONSE: Thanks — We like the figure too! We have slightly altered the illustration by increasing the size of the black dots as suggested.

REVIEWER 2

GENERAL COMMENTS:

(#17) I contest the claim that the presence of epidermal glands reflects investment in chemical signalling, and thus think that the main conclusion of this paper is overstated (although I find the results themselves nevertheless interesting). While the presence/absence of glands might reflect chemical signal investment among some closely related species (e.g., *Liolaemus*; Ruiz-Monachesi et al. 2020, Amphibia-Reptilia), skinks, snakes, varanids, and other diverse squamate groups use chemical signals (including in social interactions) and yet do not possess epidermal glands. You acknowledge exactly these points in the Discussion, which I was very pleased to see, but a reader would not get any impression of these caveats reading your title, abstract, or introduction. I think the paper (including the title) needs to be edited so that these are not conflated. See line number comments below for some specific examples.

RESPONSE: We understand the reviewer's concern (one which was also raised by the associate editor; see #1) and also appreciate that the reviewer acknowledges that we comprehensively tackle this exact concern in the discussion (L287-367). In order to inform the reader *early* in the manuscript that we are testing whether the presence of epidermal glands have impacted the evolution of lizard sociality, and that we gauge a lizard's reliance on chemical signalling by the presence/absence of these glands, we made changes —as suggested— in the title, abstract, and introduction. The title is now changed to "*Investment in chemical signalling glands facilitates the evolution of sociality in lizards*". In the abstract, on line 35-37, we added "*We used the presence of epidermal glands as a proxy of chemical investment and considered social grouping as the occurrence of social groups containing both adults and juveniles. Based on a dataset of 911 lizard species, our models strongly supported correlated evolution between social grouping and chemical signalling glands.*" In addition, we clarified in the introduction (what we already comprehensively explain in the discussion) that there are also alternative sources of semiochemicals in lizards. On line L109-117, we added the following: "*In the case of lizards that use chemical communication, several possible semiochemical sources may be responsible for signal production. While there is some evidence that cloacal exudates and faecal pellets may contain socially relevant information, the skin and the (generation and follicular) epidermal glands are considered the main sources of chemical signals in lizard communication, enabling mate assessment, individual recognition, species recognition, and sex identification (reviewed in [35–39]).*"

Liolaemus tenuis females, for instance, are more attracted to substrates covered with male epidermal gland secretions than to substrates scent-marked with male skin extracts [40].” With these changes, we believe we have made clear at the onset of the manuscript (and not only in the discussion) that the main finding of the study is that the presence of epidermal glands impact social evolution in lizard (rather than directly arguing that chemical investment impact social evolution), and that we use the presence/absence of epidermal glands as an assessment of a lizard’s reliance on chemical communication. Whether epidermal glands can be considered a good estimate of chemical signalling investment is elaborately discussed later in the manuscript. Moreover, we added the following to the discussion L322: “Other work suggests that some species rely on a combination of chemicals from different sources to enhance signal effectiveness or to broadcast different messages at once [40,88]. Skin extracts and epidermal gland secretions of male *Liolaemus tenuis* lizards, for instance, are found to mainly trigger marking behaviour in female conspecifics, whilst the scent of their faeces elicited predominantly display behaviour [40].”

(#18) This is a correlational study, as are almost all macroevolutionary studies. You are usually careful to phrase things so as not to conflate correlation with causation, with the major exception of your paper’s title, and a few smaller instances (see line comments). You cannot be certain that the presence of epidermal glands is not correlated with another trait that is directly influencing the evolution of sociality. Perhaps sociality is actually less likely to evolve in shrub-dwelling lizards (e.g., *Anolis* and chameleons), which appear less likely to possess epidermal glands (Baeckens et al. 2015). This is just the first example that came to mind; I’m not suggesting that claim is likely true. Conflating correlation and causation is, unfortunately, extremely common among macroevolutionary papers, but is nevertheless worth avoiding.

RESPONSE: We agree with the reviewer that we cannot prove causality from phylogenetic comparative studies—just as we cannot infer causality from any type of observational data (Maddison et al 2014 Syst Biol; Uyeda et al 2018 Syst Biol). However, we do not agree with the reviewer that we claim—or even imply—the existence of a causal relationship between epidermal glands and sociality. The reviewer, for instance, refers to the title, and likely to the use of the term ‘facilitates’. Yet, ‘facilitates’ does not imply causality, rather it indicates that the emergence of sociality throughout the evolutionary history of lizards occurs at a higher rate in lineages carrying epidermal glands relative to those without epidermal glands; which is exactly what our statistical analyses show. Moreover, our Markov models even

indicate that whilst the evolution of sociality depends on the presence/absence of epidermal glands, the opposite is not true: change in epidermal glands does not depend on the state of sociality ($P > 0.2$). We find the same pattern using MuSSE statistics. If we test the alternative hypothesis (social grouping facilitates the evolution of epidermal glands), we find no statistical support for that: transitions to epidermal glands happened at different rates in lineages with and without social grouping ($\Delta AIC = 2.00$, $X^2 < 0.01$, $P = 0.995$). Furthermore, transition rate to epidermal glands from a background of social grouping was significantly lower than transition rates towards social grouping from a background of epidermal glands ($\Delta AIC = 5.70$, $X^2 = 7.76$, $P = 0.005$). We added these latter MuSSE analyses to the manuscript for clarification (L182-186; L214-220). While we agree that we cannot prove causality from these tests (again: not something we aimed for), note that methods like MuSSE and Pagel's correlation test are better for providing *evidence consistent with causation* than something like a PGLS or PICs analysis, because the former models require that causes precede effects—this is not the case in purely correlation models such as PICs. We are aware that some other macro-evolutionary studies using similar methods explicitly state to have provided strong evidence for a causal connection (e.g. Halliwell et al. 2017), we do not claim such things in our manuscript. Overall, we believe we have been rather cautious with (over)statements concerning causality. We even deliberately avoid terms, such as “promotes” as it can be interpreted by some as slightly more forceful in comparison to “facilitates”.

We do agree with the reviewer that we cannot exclude that a third—unexplored—trait may have an effect on the evolution of one or both traits under study. Of course, as also acknowledged by the reviewer, this is the case for nearly every study in biology. Let's take the excellent study by Halliwell et al (2017) as an example. While they convincingly show that viviparity promotes the evolution of sociality in squamates, there are of course other variables that in turn may impact the evolution of viviparity, such as cold climate conditions (e.g. Shine 2007 Evolution), — a trait not included in the analysis. The same is true for our study. Because we fully understand that a range of other factors are in play, we avoid any claim that epidermal glands are the sole predictor of the evolution of lizard sociality. In fact, in the introduction, we explain that there are a number of factors that can influence the emergence of sociality. The current study is designed —and limited— to unravel specifically whether chemical-signalling glands may affect social evolution too. In sum, we have strong evidence of a mechanistic association between epidermal glands and sociality, but whether that is direct or mediated through confounding factors is unknown. Nevertheless, to emphasize to the reader that we, indeed, do not (claim to) provide any

causal connection, we write in the first paragraph of the discussion: *“Although one cannot infer causality from our phylogenetic comparative methods (any unconsidered traits that is correlated with the target trait could be causal [54,58,59]), our findings highlight the potential important role of chemical communication in the evolution of lizard sociality.”*

(#19) For some species with both epidermal glands and social grouping, only males possess epidermal glands (e.g., *Naultinus*, *Hoplodactylus*, *Rhacodactylus*; I assume there are more), and evidence suggests that epidermal glands in lizards do not differentiate until the onset of sexual maturity (briefly reviewed in Mayerl et al. 2015, *Amphibia-Reptilia*). Assuming I’m not mistaken about these claims, how do you reconcile these observations with your results? This study specifically tests the correlation between epidermal glands and social grouping defined as groups containing adults and juveniles. How can epidermal glands mediate interactions among adults and juveniles if juveniles do not possess functioning glands, and if (for some species) only a single sex possesses them? This is another reason why I urge caution with inferring causation, and I think these points are worth discussing.

RESPONSE: (1) The reviewer is correct in that there are species known for which only one of the sexes carries active epidermal glands (Chuahan 1986; Dujsebayaeva et al 2009; Lobo et al 2012). However, the large majority of studies simply find sexual differences in the anatomy (size) of epidermal pores, but whether this is merely a body size-effect and, more importantly, whether these pore size differences impact the secretion production, “pheromonal activity”, or signalling potential of the glands is only known for a small number of species with mixed results (Cole 1966; Alberts et al 1992, 1993; Van Wyk & Mouton 1992; Jared et al 1999; Khannoon et al 2013; Imparato et al 2007; Baeckens 2017). To show that we acknowledge that in some species there are sexual differences in gland activity, we wrote in the discussion L322: *“Other work suggests that some species rely on a combination of chemicals from different sources to enhance signal effectiveness or to broadcast different messages at once [40,85]. Skin extracts and epidermal gland secretions of male *Liolaemus tenuis* lizards, for instance, are found to mainly trigger marking behaviour in female conspecifics, whilst the scent of their faeces elicited predominantly display behaviour [40]. Exploiting multiple chemical sources may be particularly useful in lizard species for which active epidermal glands are restricted to a single sex only. In species where females lack active epidermal glands, such as *Podarcis hispanicus*, male conspecific can extract information from female cloacal and body odours to determine, for instance, reproductive condition [86], while females can profit from the information-rich epidermal gland secretions*

to assess, for example, a male's immune response [87].". (2) Similar to the aforementioned sex-differences, epidermal gland activity in juveniles has only been studied in a few species (e.g. Cole 1966; Alberts et al. 1992; Valdecantos et al. 2014). It is correct that in some lizard species, juveniles only develop active glands when they are sexually mature (e.g. *Liolaemus poecilochromus*, Valdecantos & Lobo 2007), yet most studies on the epidermal gland activity of juvenile lizards actually report that juvenile glands do produce secretions (Alberts et al. 1992; Valdecantos et al. 2014), which can convey socially relevant information (*Iguana iguana*, Alberts 1992). Because gland size is correlated with body size (Alberts 1992; Valdecantos et al. 2014; Baeckens 2017), epidermal gland size in juveniles is smaller than in adults (Alberts 1992; Valdecantos et al. 2014), but not necessarily less important for social signalling. Still, to show that we acknowledge that in some species juvenile may have inactive glands, we wrote on L334: "*Also, chemical communication in the offspring-to-parent direction may involve multi-source chemical signalling, notably in species where glandular secretions only start to differentiate at the onset of sexual maturity. In such cases, parents likely rely on a mixture of chemicals from various sources for recognizing kin [88,89]. The diversity of chemical sources in lizards and the interspecific variation in the importance of each source in producing socially relevant signals illustrates the imperfections of epidermal glands as a measure for chemical signalling investment.*"

(#20) While going through the data I found some errors. Both *Amblyrhynchus cristatus* and *Brachylophus* (2 spp.) are listed as not possessing epidermal glands, but they do, with papers published on the composition of secretions from these glands in *Amblyrhynchus* (<https://peerj.com/articles/3689/>). These species are early on alphabetically and so caught my eye easily, but it's worth checking the data more thoroughly (for *Brachylophus vitiensis*: <https://journals.plos.org/plosone/article?id=10.1371/journal.pone.0073127>).

RESPONSE: We truly appreciate the reviewer's helpful advice and apologise for the oversight. As suggested, we double-checked our data meticulously, made the necessary changes to the raw data, and re-did all the analyses after adding in these species (which did not change the overall pattern or interpretation).

(#21) I looked at the data and references from Halliwell et al., with a focus on species I'm most familiar with, just to get a personal impression of how accurate the claims of social grouping are. In your study and in Halliwell et al. 2017, social grouping is defined as: "social groups containing both adults and juveniles" (quoted from Results section of

Halliwell et al. 2017). Considering this, I take issue with the classification of *Underwoodisaurus milii* and *Christinus marmoratus* as species with social grouping. Their classification as such is based on Kearney et al. 2001 (Herpetologica), which states for *U. milii* (as *Nephrurus milii*): “For example, adult females of *N. milii* were rarely found with juvenile conspecifics”, and, “...members of an aggregation tended to resemble each other in body size”. While rarely doesn’t mean never, this does not support that this species has inter-generational social grouping. In that same study, there was evidence of aggregations in *Christinus marmoratus*, with the main finding that groups tended to contain only a single male but otherwise reflected what would be expected by chance. I’m not contesting that these species aggregate, but there’s not good evidence that there is social grouping as defined here. Considering that these species lack epidermal glands, changing their social grouping classification would make your results even stronger. But the other consequence of all this is that my confidence in the classifications made by Halliwell et al. is somewhat lower. Halliwell et al. classified many species based on a single book (Somma 2003), that for many species (e.g., *Gekko gecko*) cites mainly anecdotal observations in terraria. Because social grouping in lizards appears relatively rare, even a few misclassifications can bias the results of your statistical tests, and so it’s worth worrying about and probably checking — and at the very least, acknowledging as a limitation.

RESPONSE: We understand the reviewer’s concern. We also agree with the reviewer that the evidence for intergenerational social grouping in *Christinus marmoratus* and *Underwoodisaurus milii* is questionable after further review. Because we are of the opinion that databases should get updated when necessary, we changed the social state of both species in our dataset. Consequently, we re-did all the analyses, which did not change the overall pattern or interpretation. In our methods section, we write on L147: “For two species (*Underwoodisaurus milii* and *Christinus marmoratus*), we were made aware that they do not conform to the social classification of Halliwell et al. [13] (based on [49]) and therefore did not score them as species with intergenerational social grouping in our dataset.” We do want to assure the reviewers that also observations from captive environments are highly informative. In lizards, in the context of social grouping, social tolerance is hugely important. So, even if it is in a captive environment, if mothers tolerate their offspring, this is essential for basic parental care. To emphasize this, we added the following to the method section on L145: “Briefly, they included in their analysis any species for which there was a reported association between adults and juveniles that was indicative of social tolerance. This is

important because social tolerance is the basis for parental care in lizards and the precursor for more complex parental care [26,27].”

(#22) Why were snakes excluded from your analyses? They were included in Halliwell et al. 2017, they demonstrate social grouping as it is treated herein, and they are simply a diverse radiation of reduced-limbed lizards. Other reduced-limbed radiations such as pygopodids and amphisbaenids are also included here, so I see no sound reason to exclude snakes from the analysis. That they do not possess epidermal glands would not be a good reason because other diverse clades lacking glands (and sociality; e.g., anoles & chameleons) are included.

RESPONSE: Throughout lizard evolutionary history, generation and follicular epidermal glands independently emerged and disappeared a number of times in a range of different lizard taxa, including legless lizards. In contrast, epidermal glands have been lost in snakes and have not been re-emerged in any of the ~3,800 snake species. Yet, snakes evolved an extreme level of chemosensory specialisation for both sexual behaviour and foraging. Because of the single gland loss (and no re-emergence) and because we felt we couldn't provide enough support for the idea of epidermal glands as a proxy for chemical signal investment in snakes, we decided to focus solely on lizards in this study. Note, however, that we only interpret our data and limited our conclusions to lizards, and that we do not make any claims about the relationship between snake sociality and chemical abilities. We agree that it would be very interesting for future research to examine the hypothesis tested in this study on lizards, but by broadening the scope and including snakes, but also other (avian) reptiles and other amniotes.

(#23) According to Garcia-Roa et al. 2017, 25% of lizards have epidermal glands, but for the species included in your study, 31.7% have glands. Thus, your data are not a random sample of the phylogeny, and some groups are underrepresented (e.g., skinks). Could this sampling bias influence the results of your statistical analyses? I'm not familiar enough with the underlying assumptions of these analyses to make specific comments, but my impression is that — as with statistics generally — a non-random sample will yield spurious conclusions. If this is the case it should be discussed as a limitation of the study, and if it is not the case then I just need some reassurance. On a related note, there's contention about the validity of ancestral state reconstructions, with a recent study (Holland et al. 2020, Scientific Reports) indicating that they are often unreliable when

traits are not selectively neutral. Notably, SSE models (including MuSSE, which was used here) are prone to high type I error rates when the trait does not influence speciation and/or extinction rates. How does MuSSE perform with your data compared to other models?

RESPONSE: (1) To be honest, in our humble opinion, we believe that our sample of 911 species (~32% carrying glands) is actually a very accurate sample of the 4341 lizard species in the study by Garcia-Roa et al (2017) reporting on ~25% carrying glands, simply because the percentage difference is so minor. Regardless, one potential reason for this relatively small difference in percentage (of species carrying glands) between studies is the fact that the study by Garcia-Roa et al (2017) solely focused on follicular epidermal glands, whereas our study also incorporates species with generation glands. Another potential reason could be linked to the idea that large-scale phylogenetic analyses of character state evolution inevitably suffer from the assignment of true negatives (Cockburn 2009 Proc Biol Sci B; Freckleton 2009 J Evol Biol). Luckily though, the lower sample size of our study (N = 911) relative to the Garcia-Roa study (N = 4341) allowed us to check the data extra carefully, thereby lowering the chance of true negatives, and increasing the accuracy and validity of our collected dataset. Irrespective of our confidence in our assembled dataset, missing data and incomplete phylogenetic coverage can, indeed, result in potential biases (Nakagawa & Freckleton 2014 Syst Biol; Maddison & FitzJohn 2014 Syst Biol). On the other hand, a recent study by Marcondes (2019 PeerJ) offers a degree of reassurances in regards to the effect of missing data and incomplete phylogenetic sampling; he detected only notable bias in parameter estimation under a very high percentage (90%) of correlated missing data. Overall, our sample size, descriptive statistics, and phylogenetic comparative models do not indicate any abnormalities and should not cause any concern for its interpretation in comparison to other phylogenetic comparative studies. (2) This brings us to the reviewer's second comment. We are aware that in some cases the MuSSE models can suffer from type I errors. Note however that Holland et al (2020 Sci Rep; Holland is also co-author of the Halliwell et al. 2017 study) reports that error rates are affected by tree size and that trees with a high number of species (N = 400 and higher) show significantly lower error rates than tree with a low number of tree tips (N = 100). Nevertheless, it is because each phylogenetic comparative model bears its own string of assumptions that we examined our data using three different statistical approaches. The fact that all three approaches describe the same pattern shows and validates our interpretation.

(#24) The caption for Figure 1 indicates that branches should be coloured according to the ancestral state reconstruction, but from what I can see all branches are black.

RESPONSE: We apologise and thank the reviewer for pointing out this error in the figure caption. The caption referred to an older version of the figure, so we made the necessary changes to the text to fit the current figure.

(#25) I would like to see all R code and a file containing the pruned phylogeny added to the electronic supplementary material for scrutiny and reproducibility.

RESPONSE: Of course! All data and codes will be made publicly available upon acceptance. We would also be more than happy to share our codes with the editor and reviewer prior to acceptance if the editor deems this necessary.

MINOR SUGGESTIONS:

(#26) LN27: I suggest striking this first sentence. Your study addresses the correlates of sociality, but not *why* organisms are social, and so it seems a little out of place — plus the next sentence works well as an opener in my opinion.

RESPONSE: As suggested, we removed the following sentence: *“A key question in evolutionary biology is why animals live in stable social aggregations, such as family groups.”*

(#27) LN35: As per my earlier comment, you test the correlation with epidermal glands, not with investment in chemical signalling.

RESPONSE: As discussed in #17, we added the following to the abstract (line 35-39): *“We used the presence of epidermal glands as a proxy of chemical investment and considered social grouping as the occurrence of social groups containing both adults and juveniles. Based on a dataset of 911 lizard species, our models strongly supported correlated evolution between social grouping and chemical signalling glands.”*

(#28) LN35–37: Here you say that 32% of 910 species have epidermal glands, and *of those*, 42% are social — but this directly contradicts your raw data. In fact, only 27 of the 288 gland-bearing species are considered social here, which is 9% of species with epidermal glands.

RESPONSE: We thank the reviewer for pointing out an issue that may cause confusion. Since the correct details on the number of species with(out) social grouping and epidermal glands can be found in the first paragraph of the result section (L197-199) and in order to limited

the word count in the abstract, we removed the aforementioned statistical descriptions from the abstract.

(#29) LN40: Conflating correlation with causation. Your results highlight the “potential” importance of chemical signalling via epidermal glands. I do share your hypothesis that chemical communication is likely very important for the evolution of sociality, but these results (if robust) are correlational.

RESPONSE: As suggested, we included “*potential*” to the sentence. See #18 for a more elaborate reply on the issue of causality.

(#30) LN60–61: I’m not a fan of this opening sentence. It comes off as a bit sensationalist, like the opening of a David Attenborough documentary. Also, to be a pedant, those organisms all have other things in common, too. This is just my superficial impression—I leave the writing style entirely to you.

RESPONSE: We appreciate the reviewer’s opinion of our opening sentence. Both authors strive to write in a more engaging way and we are happy with this sentence. Given that the reviewer was happy for us to use our own writing style, we are opting to keep this sentence as written.

(#31) LN62: Stray comma. I suggest the comma goes between “sociality” and “including”.

RESPONSE: Done.

(#32) LN64: “Biologist” should be plural, “biologists”.

RESPONSE: Thanks for pointing out the typographical error. “*biologist*” is now changed to “*biologists*”.

(#33) LN90: No need for the hyphen between “short-distances”.

RESPONSE: As suggested, we removed the hyphen.

(#34) LN92: You’re not really arguing this, rather, you’re proposing it as a hypothesis.

RESPONSE: True. We therefore changed “*argue*” by “*hypothesise*”

(#35) LN96: “...for, for...”

RESPONSE: We deleted “*for*”.

(#36) LN104–107: And yet agamids overwhelmingly possess epidermal glands, while skinks lack them entirely. This calls into question LN110–112, which posits that the presence of epidermal glands reflects investment in chemical signalling, even though you’ve just mentioned that skinks (which lack epidermal glands) are more chemically oriented. I realise you qualify this with the comment about needing more empirical support, but it seems clear that the presence/absence of epidermal glands is not a good indication of “investment” in chemical signalling. I think this is a misleading way to frame the problem, and leads to unfounded conclusions when interpreting your results. See my other comments on this matter.

RESPONSE: We understand the reviewer’s concern. While our analyses show strong evidence for a role of epidermal glands in the evolution of lizard sociality, the concept of epidermal glands as a measure for chemical signalling investment is up for discussion (which is what we elaborately do in the discussion section). We do admit that our example on L104-108 was poorly chosen as it is the “exception to the rule” (we therefore changed “skinks” to “lacertids”). We appreciate that the reviewer acknowledges that although this issue does not undermine the validity or impact of our study in any way, it does require an open discussion. As written in the response on comment #1 and #17, we do this via the necessary changes in the title, abstract, introduction, and discussion.

(#37) LN113–114: There’s evidence suggesting that geckos are olfactory specialists that do not rely heavily on tongue-flicking (Schwenk 1993, J. Zoology), although I acknowledge that we need a lot more data on this.

RESPONSE: We agree with the reviewer’s comment and changed the sentence to the following: “*Lizards sample substrate-bound or airborne chemicals in the environment using tongue-flicking (vomeroolfaction), or receive chemical-laden air through the nasal nares (olfaction s.s.) [43,44].*” We also added an extra references, i.e. Schwenk 1993 (*J Zool*) on “*Are geckos olfactory specialists?*”, to back-up make this statement.

(#38) LN126: Consider adding the skink *Nangura spinosa* (sometimes *Concinnia spinosa*) to your analysis. It is in the phylogeny of Pyron et al. and has social grouping, although this species wasn’t included in Halliwell et al., possibly because the references for it (see following ref) are mainly presented in government reports and field guides (e.g., Wilson & Swan 2017). Department of Environment and Resource Management 2010. Recovery plan

for the Nangur spiny skink (*Nangura spinosa*). Report to the Department of Sustainability, Environment, Water, Population and Communities, Canberra. Department of Environment and Resource Management, Brisbane.

RESPONSE: We truly appreciate the reviewer's suggestion and input! It is great to have this rare species in our dataset. As a consequence, we re-did all of our analyses (which did not change the observed pattern or interpretation of the results).

(#39) LN127: Should be "...phylogenetic comparative analysis..."

RESPONSE: As suggested, we added "*comparative*" to the sentence.

(#40) LN131: Needs a full stop in "al."

RESPONSE: We added a ".".

(#41) LN132–135: I think the definition of social grouping used herein should be explicitly stated in the Introduction, and maybe in the Abstract if there's room for it. Social grouping can be conceptualised in various ways, so I think it's best to be as up-front about it as possible to avoid confusion (and just make it easier for the reader).

RESPONSE: We agree with the reviewer, and as suggested we included additional information on the exact definition of sociality (as used in this study) in both the abstract and introduction. One can now read in the abstract: "*Here, we take a phylogenetic comparative approach to test the hypothesis that social grouping correlates with investment in chemical signalling. We used the presence of epidermal glands as a proxy of chemical investment and considered social grouping as the occurrence of social groups containing both adults and juveniles. Based on a dataset of 911 lizard species, our models strongly supported correlated evolution between social grouping and chemical signalling glands.*" In the introduction, on L130-132, we write: "*Intergenerational social grouping (hereafter social grouping) is here defined [following 13] as the occurrence of social groupings containing both juveniles and adults.*"

(#42) LN139: A lot of the classifications regarding epidermal glands comes from Mayerl et al. 2017. Is the data from this paper publicly available? I can't find it.

RESPONSE: The data in Mayerl et al. (2015; Amphibia-Reptilia) can be found in the online supplementary material associated with the article. Unfortunately, as in so many conventional journals, one has to have a (institutional) subscription or license to access the

paper. We want to emphasize that all data and codes used in this current study will be made publicly available upon publication.

(#43) LN141: Does “epidermal glands” include generation glands, or only epidermal follicular pores?

RESPONSE: Yes. As mentioned on L152, if a species carries generation and/or follicular epidermal glands it was scored as “*epidermal glands present*”.

(#44) LN144: In the text file with your raw data, the species column states names are from Pyron et al. 2011, but I assume you actually mean Pyron et al. 2013 [ref 44]?

RESPONSE: Yes, indeed — we thank the reviewer for pointing out this typographic error. We changed “2011” to “2013”.

(#45) LN165: I assume you’re saying you used a chi-squared test? If so I suggest rephrasing to make more obvious, e.g., “We compared the likelihoods of these two models using a chi-squared test...”

RESPONSE: As suggested, we changed the sentence “*The likelihood of both models was subsequently (chi-squared) compared to test if transitions to social grouping have occurred at different rates in lineages with or without epidermal glands.*” to “*We subsequently compared the likelihood of both models using a chi-square test to examine if transitions to social grouping have occurred at different rates in lineages with or without epidermal glands.*”

(#46) LN170: Can you explain this analysis better? I had to read Halliwell et al’s methods to understand what was going on here (assuming you did this the same way they did).

RESPONSE: Sure! As suggested, we elaborated on the details of this particular statistical analysis. We have included the following: “*In the third approach, we performed ancestral state reconstructions via stochastic character mapping [54,55] to assess the strength of correlated evolution between social grouping and epidermal glands. The function ‘make.simmap’ (phytools package [52,56]) was used to fit a continuous-time reversible Markov model for the evolution of each of our two binary traits (epidermal glands and social grouping) to the tree, and to generate a set of 100 stochastic maps for each character conditioned on the model fit and tip states. The similarity of reconstructions between map sets was estimated by computing the mean and distribution of overlap in stochastic maps.*”

(#47) LN171: I'm unfamiliar with the details of make.simmap, but is there a reason the permutations are written as 100 x 100 rather than 10,000?

RESPONSE: As explained more thoroughly in #46, for each character, a set of 100 stochastic maps was generated. We formulated the calculation of its distribution and overlap by "100x100" (true, "10,000" would have been better), but this is now removed as part of the text was rewritten (see #46).

(#48) LN181: "...with respect to the phylogeny."

RESPONSE: As suggested, we included "the".

(#49) LN197: I suggest striking "highly".

RESPONSE: As suggested, we removed "highly".

(#50) LN199: Degrees of freedom are not meaningful for permutational analyses. The p-value is calculated from the permutation distribution.

RESPONSE: We rather disagree that showing the degrees of freedom is not meaningful, as we believe it informs the reader on the number of permutations used. We prefer to include the degrees of freedom, unless the reviewer and editor strongly advice not too.

(#51) LN203: Conflating correlation with causation. Remove "facilitated" and rephrase.

RESPONSE: We argue that the "facilitates" does not imply causality. See our reply to comment #18 for details on the matter.

(#52) LN203–204: Where has this pair-bonded and kin-based stuff come from? The social grouping classification is based on groups consisting of adults and juveniles as per Halliwell et al. 2017, not pair-bonded adults or anything to do with kin (except in some of their stable social grouping stuff, but that's not used here). This statement is beyond the scope of this study. Same comment goes for LN213–215. For many of the studies Halliwell et al. used to classify social grouping, there was actually no way to know whether the observed juveniles were actually the offspring of the adults they were found with, or kin.

RESPONSE: (1) For L220-231: We agree with the reviewer—this was an oversight. We changed the wording to only deal with parent-offspring relationships. *"Our phylogenetic comparative analyses indicate that the presence of epidermal glands facilitated the emergence of social groupings (parents and offspring, i.e. kin-based) throughout lizard*

evolutionary history.”. For L241: we reworded as follows “*The most obvious explanation is that chemical communication facilitates recognition and bonding among parents and their offspring.*” We acknowledge that in the absence of genetic confirmation, many cases of social grouping presented in Halliwell et al. (2017) could not be confirmed to be parents and their offspring. However, kin are expected to be in close spatial proximity following birth. Furthermore, because sociality is closely associated with viviparity [13], offspring are expected to be in close association with their mothers and in pair-bonded species, their fathers too.

(#53) LN212–225: What about the fact that for many species only males possess epidermal glands, and that juveniles of lizards possessing epidermal glands will not have fully formed and functional epidermal glands? How can epidermal glands then mediate social interactions in these cases?

RESPONSE: We refer to our response to comment #19.

(#54) LN209: It highlights the **potential role.**

RESPONSE: As suggested, we included “*potential*” in the sentence.

(#55) LN226: Must communication really be continuous?

RESPONSE: The reviewer has a good point. Therefore, we removed “*continuous*” from the sentence.

(#56) LN234: Regarding “juveniles and adults” — the cited study looks at female discrimination of “young” and “old” males, but there is no indication that these are sexually immature juveniles. In fact, the words “juvenile” and “immature” do not appear in the manuscript (except in the reference list).

RESPONSE: The reviewer is correct. Consequently, we changed “juveniles” with “young” and “adults” with “old males”.

(#57) LN262–266: These references do not demonstrate that epidermal glands are **the main source of chemical signals in lizards, even if that claim is frequently made within them. Three of these are reviews (which of course do review how epidermal glands are an **important** source of chemical signals), and one is a macroevolutionary study (although Mayerl et al. also has a bit of macro in it). None of these references test that epidermal**

glands are behaviourally more important than other sources of chemical signals (e.g., urodaeal glands, faeces, etc.). If there are studies that support this claim broadly across lizards please cite them directly. I can't help but get the impression that ref 68 is a gratuitous self-citation here.

RESPONSE: We agree with the reviewer and, as suggested, changed “*the main source*” by “*an important source*”. We removed the reference 68 to avoid any misconception on needless self-promotion. Instead, we included a reference to the work by Labra (2008) and Labra et al (2002), which report on the different sources of pheromones (in *Liolaemus*). Moreover, on L322-328, we wrote: “*Other work suggests that some species rely on a combination of chemicals from different sources to enhance signal effectiveness or to broadcast different messages at once [40,88]. Skin extracts and epidermal gland secretions of male Liolaemus tenuis lizards, for instance, are found to mainly trigger marking behaviour in female conspecifics, whilst the scent of their faeces elicited predominantly display behaviour [40].*” In addition, in the introduction on L115, we wrote “*Liolaemus tenuis females, for instance, are more attracted to substrates covered with male epidermal gland secretions than to substrates scent-marked with male skin extracts [40].*”

(#58) LN270–272: This is false. We know that skinks, varanids, and snakes rely heavily on chemical signalling, and none of these possess epidermal glands. It might be true in specific circumstances (e.g., Ruiz-Monachesi et al. 2020), but is demonstrably false in broad terms. And you go on over the rest of the paragraph to acknowledge this — there are many sources of import, socially mediating chemical signals in lizards. Thus, is there any value in stating something like: “Because of their significance in chemical signalling, the absence of epidermal glands in lizards likewise suggests less reliance on chemical signals as an information source” before diving in to these caveats?

RESPONSE: We agree with the reviewer and acknowledge that the aforementioned statement was an overgeneralization. We changed that sentence to the following: “*Because of their significance in chemical signalling, the absence of epidermal glands in lizards has been interpreted by some researchers as less reliance on chemical signals as an information source [35,40–44].*”

(#59) LN279: I suggest replacing “depauperated” with a more commonly used synonym, e.g., “reduced”.

RESPONSE: As suggested, we changed “depauperated in” with “constrained by”.

(#60) LN294–305: You say tongue-flicking is more suitable in one sentence, then in the next sentence contradict yourself and say it's too biased and flawed. You could simply say that baseline tongue-flick rates are an alternative proxy (without saying it's probably better), and then outline its strengths and weaknesses without contradicting yourself.

RESPONSE: We agree with the reviewer and deleted “, and arguably more suitable proxy” from the sentence.

(#61) LN305–309: You do not show any of this. You find a correlation between the presence of epidermal glands and the evolution of social grouping defined as associations between adults and juveniles. This is not evidence that epidermal glands facilitated the evolution of social grouping behaviour (conflating correlation and causation), and social grouping was not defined by pair-bonds or family units. Halliwell et al. give an additional and stricter classification of “stable social grouping”, but that is not what was analysed here.

RESPONSE: While we agree that our data does not provide information on causality, we do not agree that the use of “facilitates” implies a causal relationship. For details on the issue of causality see our response to comment #18. In regards to the complex sociality comment, we agree with the reviewer that we did not explicitly test this. Therefore, we removed the following from the sentence: “*but also pair bonding and family unit (complex sociality)*”.

(#62) LN327: Branch colours are not visible on the figure. All branches are black.

RESPONSE: We thank the reviewer for pointing out this error in the figure caption. The caption referred to an older version of the figure, so we made the necessary changes to the text to fit the current figure.

General comment:

(#63) The authors are suggesting that the mode of communication facilitates the occurrence of sociality, i.e that the initial presence of glands allows and contributes to the development of sociality. However, in principle, one would expect that signal complexity increases as the need for communication increases. Or more likely that there is a coevolution process in which an initial low investment in communication increases to a higher level of investment as sociality develops. I was not convinced that these alternative scenarios were properly refuted or even discussed.

RESPONSE: Indeed —in our study, we test whether the reliance upon chemical signaling (using the presence of epidermal glands as a proxy) has facilitated the emergence of social aggregations in lizards. The results of our phylogenetic comparative analyses provide strong support for this idea. Although our study was designed to challenge this specific research question, we agree with the reviewer that an alternative and interesting question is whether chemical signal complexity (as a measure for message diversity) correlates with social complexity. Most tests of the so-called “social complexity hypothesis for communicative complexity” (Ord & Garcia-Porta 2012 *Phil Trans R Soc B*; Peckre et al 2019 *Behav Ecol Sociobiol*) has focused on visual and acoustic signals (Peckre et al 2019 *Behav Ecol Sociobiol*), with only a few excellent studies tackling chemical signals, e.g. in insects (Wittwer et al 2017 *PNAS*) and in primates (delBarco et al. 2012 *Phil Trans R Soc B*). Of course, such a comprehensive comparative analysis of co-evolution requires detailed information on the composition, diversity, and richness of the chemical signals of a large number of species with varying degrees of social organization. Unfortunately, and likely because sampling and chemical analyses are still so labour-intensive, chemical signal design has only been addressed for a few lizard taxa only (Martín & López 2014 in *Reproductive Biology and Phylogeny of Lizards and Tuatara*; Mayerl et al 2015 *Amphibia-Reptilia*; Baeckens et al 2019 *Belg J Zool*), covering little variation in the degree of lizard social organization. This poor and clustered phylogenetic sampling of lizard chemical signals severely constrains the analyses of co-evolution between chemical signal complexity and social complexity in lizards at this point in time. Irrespective of the fact that our study was designed to challenge a different research question, we do agree with the reviewer that the topic of signal complexity vs. social complexity deserves more attention in the manuscript. Therefore, we added the following paragraph to the discussion (L351-360): *“Ideally, one would possess detailed*

information on the composition, diversity, and richness of species' chemical signals [as in 87,88]. This would allow tests of co-evolution between signal complexity and the level of social organisation ["social complexity hypothesis for communicative complexity"; 89,90] as shown in halictid bees [91] and strepsirrhine primates [92,93]. Alas, most of our (limited) knowledge of lizard chemical signal design originates from a few lizard taxa only [reviewed in 71], covering little variation in the degree of social organisation; such clustered phylogenetic sampling constrains, at this point, reliable co-evolutionary diversification analyses of chemical signal and social complexity in lizards."

Minor comments

(#64) Abstract: I find this sentence in the Abstract confusing: "We found that roughly 32% of 910 species invest in signal-producing epidermal glands and of these, approximately 42% form social groups." This seems to imply that 58% of species with epidermal glands, i.e. a majority of them, do not form social groups, and this seems to contradict your main conclusion. People reading the abstract without having read the rest of the paper will get confused.

RESPONSE: We thank the reviewer for pointing out an issue that may cause confusion. Since the correct details on number of species with(out) social grouping and epidermal glands can be found in the first paragraph of the result section, and in order to limited the word count in the abstract, we removed the aforementioned statistical descriptions from the abstract.

(#65) Lines 81-82: "Whether the mode of communication has influenced social evolution in terrestrial vertebrates remains largely unexplored." No references are given here, which overemphasizes the "unexplored" claim to a point that is not correct. For example, there are studies in strepsirrhine primates on the matter.

RESPONSE: We understand the reviewer's concern and apologize for neglecting the work on strepsirrhine primates. With (i) the addition of "*To the best of our knowledge*", with (ii) the adverb "*largely*" pointing out that while much still remains unexplored, some work has tackled this particular question, and (iii) with the inclusion of a reference (i.e. Drea 2020 Phil Trans R Soc B) indicating that the limited work has focused on strepsirrhine primates, we believe to have improved the aforementioned sentence.

(#66) Line 96: correct "for, for"

RESPONSE: Done.

Appendix B

Dear Dr. Rowe,

On behalf of my co-author and I, thank you very much for accepting our manuscript, entitled **“Investment in chemical signalling glands facilitates the evolution of sociality in lizards”**, for publication in *Proceedings B*. The associate editor and reviewer 2 advised to make some minor textual changes before uploading the final files for publication. We have done our best to amend the text in the ways they suggest. As suggested by the associate editor, we paid particular attention to the recognition and discussion of the study limitations.

Again, we appreciate the editor and reviewers’ helpful comments, which we believe have significantly increased the quality of the manuscript.

Sincerely,

Simon Baeckens

AUTHORS' RESPONSE TO THE REVIEWERS' COMMENTSASSOCIATE EDITOR

Comment (#1): The authors did a good job and the manuscript improved through the revision process. I agree with the referee that including snakes would make the study much stronger, and therefore I think the authors should consider to include snakes. However, I see the argument of the authors and I think the exclusion of snakes is not fatal. Nevertheless, the authors should address the other comments of the referee and add some sentences about the limitations of the data.

Response: We thank the associate editor for acknowledging the substantial efforts we have made in revising our manuscript and responding to the reviewer's feedback. In regards to the suggestion to include data on snake sociality, well, we agree that adding extra data would make any study stronger. Although we considered incorporating snake sociality data to our study, we strongly feel it would exceed the study aims and scope and would complicate the interpretation of the data to lizard biology. We do want to stress that we only interpret our data and restrict our conclusions to lizards, and that we do not make any claims about the relationships between sociality and chemical abilities in snakes. We do appreciate that the editors value the manuscript as it stands and agree with its publication with data covering lizards only. Following the associate editor's advice, we focused on addressing the other comments of the referee.

REVIEWER 2

Comment (#2): As I stated in the previous review, I am uneasy about the overall lack of rigour stemming from the use of the data from Halliwell et al. I acknowledge that the available information is extremely limited; accordingly, Halliwell et al. were necessarily not very strict in what warranted evidence of social grouping (see ‘Assigning social grouping’ section of their Methods). Furthermore, scrutiny of sources was relaxed because much of the literature was unavailable: from the methods of Halliwell et al.: “[We]... relied on the interpretation of authors when we could not access primary sources”. I am uneasy when data are inscrutable, and using such data can perpetuate errors from publication to publication. But I also recognise that one must do what they can with what’s available. As such, I’d like to see more discussion of the limitations stemming from all this. This brings me to a related point. The underlying data reflect a whole assortment of sources, some of which are detailed studies, but many of which are anecdotal observation from the field or of captive animals. Detailed studies are obviously more reliable than anecdote. For example, if any single observation of social tolerance warrants being classified as social grouping, *Underwoodisaurus milii* and *Christinus marmoratus* would indeed be classified as such (contrary to my comments in the last review and your subsequent edits). But more detailed evidence suggests that, at least in *U. milii*, aggregations of young and adults are generally avoided — but such aggregations are still sometimes observed! Thus, there is a mismatch between social classification based on anecdote, and social classification based on more rigorously collected data. My concern is that many species are misclassified based on anecdote. I admit that I’m not sure what practical steps you can take to address this here, other than perhaps reviewing the literature yourself and pruning those species without sufficient evidence, which might leave with very little left to analyse.

RESPONSE: We understand the reviewer’s concern and agree that the study by Halliwell *et al.* has its limitations. We also agree with the reviewer that —although it has its limitations— the data amassed by Halliwell *et al.* is still the most comprehensive collection on lizard sociality to date. We also agree with the reviewer’s statement that “one must do what they can with what’s available”, which is (in our humble opinion) still a lot in our case. Although we feel that our manuscript is not the most appropriate place to elaborate in detail on the limitations of the study by Halliwell *et al.*, we do agree that we have to acknowledge this. As suggested, we implemented the following on L152-156: “A small part of the data collected

by Halliwell et al. [13] was not retrieved from primarily literature, but from reports of observations by trained herpetologists. While we recognize its limitations, the data amassed by Halliwell et al. [13] is the most exhaustive dataset on lizard sociality to date.”

Minor comments:

Comment (#3): From your response: “Yet, ‘facilitates’ does not imply causality, rather it indicates that the emergence of sociality throughout the evolutionary history of lizards occurs at a higher rate in lineages carrying epidermal glands relative to those without epidermal glands”. ‘Facilitates’ does in fact imply a degree of causality. Facilitate is defined as a verb that “makes an action or process easier.” To make something easier, it must influence (causally) the outcome of the action or process. You could say, however, that the association between glands and grouping suggests that glands might facilitate grouping. I simply suggest qualifying statements such as the one on line 362 to better reflect the uncertainty surrounding correlational analyses. As I stated in the last review, in most cases you are careful not to conflate correlation with causation; I am just being nit-picky about the few exceptions.

RESPONSE: We appreciate the acknowledgment of the reviewer in that we are overall careful with not conflating correlation with causation. With the following sentence in the first paragraph of the discussion, we believe we are very explicit with emphasizing that we do not infer causality from our results in any way: *“Although one cannot infer causality from our methods (any unconsidered traits that is correlated with the target trait could be causal [55,59,60]), our findings highlight the potential important role of chemical communication in the evolution of lizard sociality.”*

Comment (#4): Regarding the ancestral state reconstructions via stochastic character mapping, you respond: “We rather disagree that showing the degrees of freedom is not meaningful, as we believe it informs the reader on the number of permutations used. We prefer to include the degrees of freedom, unless the reviewer and editor strongly advice not too.” Ok I think I was confused here. I was previously under the false impression that this was a permutation test (in which there are no ‘degrees of freedom’ in the sense of null-hypothesis significance testing), but I can see now that’s not the case. Although, I admit I’m still a bit confused about this analysis even with the additional details you’ve given. Specifically, how is the hypothesis test being done? That you report a t-score suggests a t-test. You state: “The similarity of reconstructions between map sets was estimated by

computing the mean and distribution of overlap in stochastic maps.” My impression isn’t that you tested for a difference between the map sets, but rather that you tested for a difference between the distribution of overlap in the map sets against an expectation of 0.5 (is this correct?). Can you add detail explicitly stating how the hypothesis test was done? I hope I’m not just misinterpreting everything; and I apologise if I am.

RESPONSE: No need to apologise at all! The reviewer is correct in that we test for a difference between the distribution of overlap in the map sets against an expectation of 0.5. To make this clearer in the material and methods section, we added the following this: *“A t-test was used to assess the difference between the distribution of overlap in the map sets against an expectation of 0.5.”*

Comment (#5): Your response here hasn’t addressed my concern: How do you reconcile your interpretation that glands might better facilitate social grouping with the observation that, in some of these species, females lack glands? I acknowledge that in many species the glands of females are smaller rather than entirely absent — but the exceptions matter if they are also social species. Of the 27 species with glands and social grouping, how many species have females lacking these glands? If it’s a very small number, then it’s probably still consistent with your results; but if it’s closer to half then it seems equivocal. (Off the top of my head I can see ~10 out of 27 spp. for which I think females lack functioning glands, but I could very well be wrong.) If glands do better facilitate the evolution of social grouping and mediate social tolerance among adults and young, then we’d expect both sexes to possess them. You go on to explain how there are potentially other sources of chemical signals that could mediate interactions, but this is only tangentially relevant to my comment, which specifically concerns your results for epidermal glands, and not chemical signalling investment more broadly. Ok, I realise this is getting long for a ‘minor comment’... I recommend that, instead of trying to explain away this (potential) inconsistency by invoking other chemical sources to fit it into your hypothesis, you simply 1) state that glands can be absent in females, and state the number of species with social grouping in which this is the case; 2) discuss whether this number is small or large, and thus how consistent it is with your hypothesis; and 3) if it is inconsistent, highlight how interesting it is, and what it might mean for future research that tries to better understand chemically-mediated social tolerance and sociality.

RESPONSE: We understand the reviewer’s concern. Unfortunately, the presence or lack of active glands in female lizards is not well documented in the literature at all. As in many

studies (Zucker and Beery 2010, Nature), there seem to be a literature bias towards male reports of lizard gland biology. In cases where female anatomy is considered, it is often unclear whether researchers refer to absence/presence of pores (i.e. the pore-like scales in the femoral region), glands, or glands that are active. Because of this discrepancy in reporting, we feel uncomfortable with pinpointing a specific value on the number of social lizards with females lacking glands; in our dataset, we believe that the number can vary between 6 or 10 species depending on the source. A much more rigorous examination (with preserved or preferable live animals) is preferred for female specimens. Regardless, we fully agree with the reviewer that it would be highly interesting for future research to focus on those social species for which only one of the sexes is known to carry active epidermal glands. Therefore, we added the following to the discussion: *“It would be interesting for future studies to examine chemically-mediated social tolerance among adults and young in species with females lacking active epidermal glands.”*. We also hypothesize about the potential scenarios: *“Exploiting multiple chemical sources may be particularly useful in lizard species for which active epidermal glands are restricted to a single sex only. In species where females lack active epidermal glands, such as Podarcis hispanicus, male conspecifics can extract information from female cloacal and body odours to determine, for instance, reproductive condition [86], while females can profit from the information-rich epidermal gland secretions to assess, for example, a male’s immune response [90].”*. Because it is important to underscore the limitations of the study, as underscored by the reviewer, the respective paragraph ends with the following: *“The diversity of chemical sources involved in lizard communication and the inter- and intraspecific variation in the importance of each source in producing socially relevant signals illustrates the imperfections of epidermal glands as a measure for chemical signalling investment.”*

Comment (#6) LN113: I still contest the claim that epidermal glands are the “main source” — again, skinks and snakes (which are lizards, regardless of whether they’ve been included herein) demonstrate this. More accurately, you could say that glands are likely the main source in species that possess these glands, which is what the study you summarise on *Liolaemus tenuis* actually supports. As far as I’m aware, the claim has never been tested among squamates more broadly.

RESPONSE: In order to solve this issue, we changed “the main sources” by “important sources”.

Comment (#7) LN128–129: Would you consider social grouping as defined here to be “complex sociality”? To me, it seems to be close to the simplest end of the simple–complex spectrum of sociality.

RESPONSE: We agree with the reviewer and removed “complex” from the sentence.

Comment (#8) LN213–214: Are there confidence intervals for these rate estimates? I assume there must be some degree of uncertainty as a result of uncertainty in the ancestral state reconstructions.

RESPONSE: Rate estimates were attained from the MuSSE tests, not the ancestral state reconstruction, and do not provide uncertainty values. For more information on the multistate speciation and extinction models, we refer to FitzJohn 2012 (Methods in Ecology and Evolution).

Comment (#9) LN230: I suggest either replacing i.e. with e.g., or else entirely removing “kin-based”. For many species classified as social grouping, there was no information whether grouping was between kin. It seems more likely that’s the case, but it nevertheless remains an untested assumption for many species.

RESPONSE: We agree with the reviewer and removed “i.e. kin-based” from the sentence.

Comment (#10) LN245: Suggest rephrasing: “...with their mothers, and potentially also their fathers in pair-bonded species.” Or something along those lines — the previous wording was confusing when I first read it.

RESPONSE: We thank the reviewer for this suggestion. We changed the respective part of the sentence to “... offspring are expected to be in close association with their mothers, and potentially also their fathers in pair-bonded species.”

Comment (#11) LN309: Full-stop missing at end of sentence (after the citations).

RESPONSE: Thanks for pointing out this typo. We included a full stop.

Comment (#12) LN309 & 313: *Egernia* should be italicised.

RESPONSE: We italicised *Egernia*.

Comment (#13) LN287–366: This is a very long paragraph. I suggest breaking it up into 2–3 paragraphs if it's logical to do so.

RESPONSE: That is true. We ended up dividing it into 3 smaller paragraphs.